# Correcting Split Selection in Online Decision Trees via Anytime-Valid Inference

**Salim I. Amoukou** [1]   **Saumitra Mishra** [1]   **Manuela Veloso** [1]

## Abstract

Bagging-based ensembles, most notably Adaptive Random Forests, are among the strongest performers for learning from data streams. A common denominator across these methods is their reliance on Hoeffding Trees as base learners, which grow decision trees incrementally by testing whether a candidate split is significantly better than its alternatives using concentration inequalities. Despite their empirical success, existing variants lack valid statistical guarantees. Current analyses rely on fixed-sample concentration bounds, while split decisions are made using data-dependent stopping rules, which invalidates their guarantees and can drive the probabilty of incorrect splits to one. We introduce a principled alternative based on *anytime-valid inference*. Our method provides: (i) anytime-valid control of false splits under arbitrary data streams, including non-stationary settings; (ii) finite commitment time under a predictive advantage; and (iii) under stationary i.i.d. data, risk is monotone decreasing and strictly improves at every split. Empirically, we evaluate both standalone trees and their use within Adaptive Random Forests on non-stationary streams. Our method improves performance while producing substantially smaller trees.

## 1. Introduction

In streaming environments, where data arrive sequentially and models must be updated online, ensemble methods have emerged as the dominant paradigm. In particular, bagging-based approaches such as *Adaptive Random Forests* (ARF) (Gomes et al., 2018), along with related variants including Streaming Random Patches (Gomes et al., 2019), and K-Nearest Leaves (Sun et al., 2022), consistently rank among the top performers on recent benchmarks (Montiel et al., 2020; Wang et al., 2022; Aspis et al., 2025). Among

these, ARF has effectively become the *de facto* standard for ensemble learning in data streams, across both classification (Gomes et al., 2017) and regression (Gomes et al., 2018).

A striking commonality across these methods is their universal reliance on the Hoeffding Tree (HT), also known as the Very Fast Decision Tree (VFDT) (Domingos & Hulten, 2000), as the fundamental base learner. Despite its empirical success and wide adoption, the statistical foundation of the Hoeffding Tree is not valid. The HT's core idea is to grow tree incrementally: at each node, it tests if sufficient data has arrived to commit to a split. The original algorithm invalidly applied Hoeffding's inequality to non-linear impurity measures (e.g., Gini index). Subsequent corrections either substituted linear measures (e.g., misclassification error) (Matuszyk et al., 2013) or applied more general bounds like McDiarmid's inequality (Rutkowski et al., 2012; De Rosa & Cesa-Bianchi, 2017; Jaworski et al., 2017).

We argue that these fixes remain statistically invalid. All existing approaches overlook a fundamental issue: *standard concentration inequalities assume a fixed sample size, yet the HT algorithm employs a data-dependent stopping rule.* This phenomenon, known as *optional continuation* (Grünwald et al., 2020; Shafer, 2024), invalidates the guarantees of fixed-time concentration inequality. In particular, the probability of a false split can inflate to one (see Fig. 1 in (Howard et al., 2021)).

To resolve this, we abandon fixed-sample-size inequalities and instead leverage recent advances in sequential testing, specifically the *safe, anytime-valid inference (SAVI)* framework (Shafer & Vovk, 2001; Shafer et al., 2011; Hendriks, 2018; Jun & Orabona, 2019; Orabona & Jun, 2023; Waudby-Smith & Ramdas, 2024; Ramdas & Wang, 2025). This paradigm is inherently robust to optional stopping.

**Our main contributions are:**

1. We derive a split criterion that remains valid under arbitrary data-dependent stopping rules. Our method controls false split decisions simultaneously over the entire lifetime of the tree, and applies to non-stationary and dependent data streams. When a candidate split exhibits a persistent predictive advantage, our procedure commits to the split in finite time.

2. Under stationarity, the learned tree's expected loss is

---

[1]J.P. Morgan AI Research. Correspondence to: Salim I. Amoukou <salim.ibrahimamoukou@jpmorgan.com>.

*Proceedings of the 43rd International Conference on Machine Learning*, Seoul, South Korea. PMLR 306, 2026. Copyright 2026 by the author(s).

non-increasing over time, and improves monotonically as data accumulate.

3. We validate our method across several non-stationary environments, both as a standalone replacement for Hoeffding Trees and as a drop-in substitute within Adaptive Random Forests. The resulting ensemble of trees achieve higher predictive performance while being consistently smaller in depth or number of nodes.

## 2. Problem Definition

Let $(\mathcal{F}_t)_{t \geq 0}$ be the filtration representing all information available up to time $t$, including the data and any internal randomization: $\mathcal{F}_t := \sigma((X_1, Y_1), \ldots, (X_t, Y_t), U)$ where $U$ denotes algorithmic randomness (if any). A quantity chosen at time $t$ is *predictable* if it is $\mathcal{F}_{t-1}$-measurable.

For clarity, we present the problem for classification decision trees with binary splits on continuous features; extensions to multiway splits, categorical features, and regression follow by standard adaptations (Rutkowski et al., 2020). Each leaf node update mechanism has two components: (i) a *split measure* quantifying split quality, and (ii) a *split condition* determining whether the node should be partitioned.

A decision tree recursively partitions the input space $\mathcal{X} \subseteq \mathbb{R}^d$ into a collection of axis-aligned regions. Each node $v$ corresponds to a measurable region $R(v) \subseteq \mathcal{X}$. A split candidate is a pair $c = (j, s)$ with feature index $j \in [d]$ and threshold $s \in \mathbb{R}$, inducing child regions

$$R(v_L^c) = R(v) \cap \{x_j \leq s\}, \qquad R(v_R^c) = R(v) \cap \{x_j > s\}.$$

At each node $v$, the set of admissible splits is restricted to a finite candidate class $C_v \subseteq \{1, \ldots, d\} \times \mathbb{R}$. The finiteness of $C_v$ reflects practical implementations (e.g., discretised thresholds, quantiles) and ensures that optimality is always defined with respect to a fixed, finite hypothesis class.

**Batch decision tree learning.** Given a fixed dataset $\mathcal{D}_t = \{(X_i, Y_i)\}_{i=1}^t$, an effective split maximally reduces node impurity. Define the samples reaching node $v$ as $A_t(v) := \{(X_i, Y_i) \in \mathcal{D}_t : X_i \in R(v)\}$, $n_t(v) := |A_t(v)|$. Let $p_t(v)$ denote the empirical class proportions, with components $p_{t,k}(v) = \frac{1}{n_t(v)} \sum_{(X_i, Y_i) \in A_t(v)} \mathbb{1}\{Y_i = k\}$. Common impurity measures $\mathcal{I}(\cdot)$ include entropy and the Gini index: $\mathcal{I}_{\text{entropy}}(p) = -\sum_k p_k \log \mathbf{p}_k$, and $\mathcal{I}_{\text{Gini}}(p) = \sum_k p_k(1 - p_k)$. For a candidate split $c = (j, s) \in C_v$, the empirical impurity decrease at node $v$ is

$$\Delta_t^{v,c} = \mathcal{I}(\mathbf{p}_t(v)) - P_{t,L}\,\mathcal{I}(\mathbf{p}_t(v_c^L)) - P_{t,R}\,\mathcal{I}(\mathbf{p}_t(v_c^R)),$$

where $P_{t,L} = n_t(v_c^L)/n_t(v)$, and $P_{t,R} = n_t(v_c^R)|/n_t(v)$.

In batch learning, the split at node $v$ is chosen by maximising the empirical impurity reduction over the candidate set $c^\star = \arg\max_{c \in C_v} \Delta_t^{v,c}$.

**Online decision tree learning.** In a streaming setting, observations $(X_t, Y_t)_{t \geq 1}$ arrive sequentially and the dataset is never fully observed. The batch procedure is therefore inapplicable; instead, the learner must decide *online* when sufficient evidence justifies a split.

For each node $v$ and split $c \in C_v$, define the *population impurity decrease* as

$$\Delta^{v,c} = \mathcal{I}(p(v)) - P_L\,\mathcal{I}(p(v_c^L)) - P_R\,\mathcal{I}(p(v_c^R)), \quad (1)$$

where probabilities correspond to their population counterparts under the data-generating distribution. Ideally, the learner would select a split $c^\star$ satisfying

$$\Delta^{v,c^\star} > \Delta^{v,c'} \qquad \forall c' \in C_v, \qquad (2)$$

that is, the empirically chosen split coincides with the population-optimal one over the same finite candidate class.

Since $\Delta^{v,c}$ is unknown, online algorithms rely on statistical inference from empirical estimates $\Delta_t^{v,c}$ computed from the data observed so far. A common strategy introduces a threshold $\varepsilon(n_t(v), \delta)$ such that, if

$$\Delta_t^{v,c^\star} - \Delta_t^{v,c'} > \varepsilon(n_t(v), \delta) \quad \forall c' \in C_v,$$

then, with probability at least $1 - \delta$, the selected split is population-optimal.

Hoeffding Trees (Domingos & Hulten, 2000) instantiate $\varepsilon(\cdot)$ using Hoeffding's inequality, despite the non-linearity of impurity criteria. Subsequent work replaces this with McDiarmid-type bounds (Rutkowski et al., 2012; De Rosa & Cesa-Bianchi, 2017; Jaworski et al., 2017) or linearized impurity measures (Matuszyk et al., 2013).

**The overlooked issue.** All existing guarantees are valid given a *fixed sample size*. In practice, however, impurity estimates are updated sequentially, and the algorithm commits to a split at the *first* time the stopping condition is satisfied. This induces a data-dependent stopping time that depends on the entire data trajectory.

Let $(C_t)_{t \geq 1}$ be a sequence of confidence intervals for a parameter $\delta$, constructed using a classical concentration inequality (e.g., Hoeffding's inequality). Such bounds provide only *fixed-time* control: for any fixed $t$,

$$\mathbb{P}(\delta \notin C_t) \leq \alpha,$$

which offers no guarantee under data-dependent stopping. Instead, we require *anytime-valid* coverage:

$$\mathbb{P}(\exists t \geq 1 : \delta \notin C_t) \leq \alpha, \quad \text{or} \quad \mathbb{P}(\forall t \geq 1 : \delta \in C_t) \geq 1 - \alpha.$$

This immediately implies validity at any (possibly infinite) stopping time $\tau$:

$$\mathbb{P}(\delta \notin C_\tau) \leq \alpha, \qquad \mathbb{P}(\delta \in C_\tau) \geq 1 - \alpha.$$

Standard bounds fail under optional stopping (see Fig. 1 in (Howard et al., 2021)) and therefore do *not* control the probability of selecting an incorrect split at the time the algorithm actually stops.

Moreover, a fundamental gap exists between theory and practice. Classical Hoeffding-based analyses assume independent observations, yet Hoeffding Trees are most often deployed inside ensemble methods such as Adaptive Random Forests, which operate on non-stationary and potentially dependent streams. This directly violates the assumptions underlying existing guarantees. These observations motivate our central question:

*How can we design a splitting criterion that remains valid under data-dependent stopping and non-stationary, dependent data streams?*

## 3. Valid Sequential Tests for Split Selection

Our approach departs from the classical objective of selecting the split that maximizes population impurity reduction (cf. (2)). In non-stationary environments, the population impurity (1) may itself be ill-defined, since the data-generating distribution may evolve over time and the optimal split may change accordingly. Rather than comparing candidate splits *against each other* under a static distribution, we instead ask a different question:

*Is a proposed split demonstrably better than leaving the node unsplit, based on the data observed so far?*

This reframes split selection as an *online model comparison* problem. Specifically, we compare two predictors: an *incumbent* model corresponding to the unsplit leaf, and a *challenger* model obtained by applying a candidate split. The goal is to determine, using accumulated information up to time $t$, whether the challenger has achieved better predictive performance than the incumbent under potentially non-stationary and dependent data.

Predictive performance is evaluated using a loss function $\ell$, with log loss and Brier score serving as loss-based analogues of entropy and Gini impurity. Our theoretical analysis assumes bounded losses $\ell \in [0, 1]$; when this condition does not hold in practice, we enforce it via normalization.

### 3.1. Testing a Candidate Split

Fix a leaf node $v$ created at time $s^v$. Until a split is committed, node $v$ acts as the *incumbent* predictor.

At time $s^v$, the learner selects a candidate split $c \in C_v$. This defines a hypothetical *challenger* model that partitions $R(v)$ into two child regions. No structural changes are made at this stage: the challenger is evaluated in *shadow mode* and does not affect future routing decisions.

- Incumbent $m^v$ predicts using the empirical label distribution at node $v$, estimated from data observed up to the current time.

- Challenger $m^{v_c}$ identical to the incumbent, except that node $v$ is split according to $c$, with each child maintaining its own empirical label distribution.

For each $t \geq s^v + 1$,

1. Both predictors output *prequential* predictions based on $\mathcal{F}_{t-1}$.

$$m_{t-1}^v(x) = p_{t-1}(v), \qquad x \in R(v),$$

$$m_{t-1}^{v_c}(x) = \begin{cases} p_{t-1}(v_c^L), & x \in R(v_c^L), \\ p_{t-1}(v_c^R), & x \in R(v_c^R), \end{cases} \quad (3)$$

2. After observing $Y_t$, we compute the losses and define the bounded difference

$$\Delta_t^{v,c} := \ell\big(m_{t-1}^v(X_t), Y_t\big) - \ell\big(m_{t-1}^{v_c}(X_t), Y_t\big) \in [-1, 1].$$

3. Both predictors are updated identically using $(X_t, Y_t)$; their only distinction remains the proposed split.

Since both predictions are $\mathcal{F}_{t-1}$-measurable, the conditional mean is well defined:

$$\delta_t^{v,c} := \mathbb{E}[\Delta_t^{v,c} \mid \mathcal{F}_{t-1}].$$

**Defining improvement.** In non-stationary environments, there are two standard hypothesis-testing formulations for comparing predictive models over time (Lehmann et al., 1975; Rosenbaum, 1995; Ehm & Krüger, 2018; Choe & Ramdas, 2024).

The first formulation tests whether a challenger ever attains an advantage over the incumbent. This corresponds to the *strong null hypothesis* (Choe & Ramdas, 2024):

$$H_0^{v,c}: \qquad \forall t \geq s^v + 1, \quad \delta_t^{v,c} \leq 0.$$

Under this hypothesis, the challenger is assumed to never outperform the incumbent at any point in time.

The second formulation evaluates whether the challenger improves upon the incumbent on average over time. This leads to the *weak null hypothesis*:

$$H_{\text{w},0}^{v,c}: \qquad \forall t \geq s^v + 1, \quad \frac{1}{t - s^v} \sum_{u=s^v+1}^{t} \delta_u^{v,c} \leq 0.$$

The choice between these formulations depends on the specific application. Recent work (Choe & Ramdas, 2024) suggests using the strong null when comparing closely related algorithms operating on the same data stream, and the weak null when comparing substantially different models.

**Choice in our setting.** We adopt the *strong* null $(H_0^{v,c})$ as our primary testing target. This aligns with the recommendation of (Choe & Ramdas, 2024) to use the strong null when comparing closely related algorithms trained on the same data. Indeed, during the test period, both models are trained using the same data and are directly related by construction. Empirically, we also found that the *weak* null leads to more conservative behaviour: splits are committed later, resulting in inferior predictive performance. This is expected, since the weak formulation relies on averaged effects that can be influenced by accumulated history.

For completeness, we derive sequential tests for both hypotheses. For notational brevity, we omit the superscripts $(v, c)$ where the context is unambiguous.

## 3.2. Anytime-Valid Test via Testing by Betting

We adopt the *testing-by-betting* framework (Shafer & Vovk, 2001; Shafer et al., 2011; Hendriks, 2018; Jun & Orabona, 2019; Orabona & Jun, 2023; Waudby-Smith & Ramdas, 2024; Ramdas & Wang, 2025), which interprets statistical hypothesis testing as a *betting game against the null hypothesis*. At each time step, the bettor wagers on whether the challenger outperforms the incumbent, as measured by the sign of $\Delta_t$. Under the null hypothesis $H_0$, no betting strategy can achieve positive expected growth. Under the alternative, the bettor's wealth increases over time.

**Testing by betting.** The bettor starts with unit wealth $W_s = 1$. At each time $t \geq s + 1$, a predictable betting fraction $\beta_t \in [0, 1]$ is chosen based on $\mathcal{F}_{t-1}$. After observing $\Delta_t$, wealth is updated according to

$$W_t = W_{t-1}\big(1 + \beta_t \Delta_t\big). \tag{4}$$

Since under the *strong* null $H_0$ we have $\mathbb{E}[\Delta_t \mid \mathcal{F}_{t-1}] \leq 0$, the process $(W_t)_{t \geq s}$ is a nonnegative supermartingale.

By Ville's inequality, for any $\alpha \in (0, 1)$,

$$\mathbb{P}_{H_0}\left(\sup_{t \geq s} W_t \geq \frac{1}{\alpha}\right) \leq \alpha.$$

Consequently, thresholding the wealth process yields an anytime-valid test: regardless of the betting strategy, under $H_0$ the wealth exceeds $1/\alpha$ with probability at most $\alpha$. Therefore, a wealth above this threshold is a strong evidence that the null is false.

**Portfolio interpretation.** Following (Orabona & Jun, 2023), we can equivalently represent the betting game as an online portfolio selection with two stocks. Define two assets with returns

$$R_t^{(0)} := 1, \qquad R_t^{(1)} := 1 + \Delta_t \in [0, 2].$$

Investing a fraction $\beta_t$ in asset 1 and $1 - \beta_t$ in asset 0 yields gross return

$$(1 - \beta_t)R_t^{(0)} + \beta_t R_t^{(1)} = 1 + \beta_t \Delta_t,$$

which coincides with (4) one step's update. Designing a powerful anytime-valid test therefore reduces to selecting an effective online portfolio strategy.

Rather than tuning $\beta_t$, we adopt a parameter-free strategy: the *Universal Portfolio* (UP). UP is minimax-optimal with respect to the best constant rebalanced portfolio and, in i.i.d. markets, achieves the optimal growth rate of any fixed strategy (Cover & Thomas, 1998). UP constructs a mixture over constant rebalanced portfolios using a Jeffreys prior.

Concretely, we define

$$\beta_t := \frac{\int_0^1 \beta \prod_{u=s}^{t-1}\big(1 + \beta\, \Delta_u\big)\, dF_+(\beta)}{\int_0^1 \prod_{u=s}^{t-1}\big(1 + \beta\, \Delta_u\big)\, dF_+(\beta)}, \quad F_+ = \text{Beta}\big(\tfrac{1}{2}, \tfrac{1}{2}\big),$$
$$\tag{5}$$

and update wealth via (4). The resulting process remains a nonnegative supermartingale under $H_0$ and empirically accumulates evidence rapidly when the challenger is superior.

**Replacement rule and global error control.** We run one such test for each candidate split available at a leaf. Given per-test significance levels $\alpha^{v,c}$, we replace the incumbent with the challenger at the stopping time

$$\tau^{v,c} := \inf\left\{t \geq s^v + 1 : W_t^{v,c} \geq \frac{1}{\alpha^{v,c}}\right\}.$$

We show in the next section that it yields *global* control: with probability at least $1 - \alpha$, no false replacement ever occurs over the entire stream. When multiple candidates cross their thresholds simultaneously, we select the one with the largest wealth.

## 3.3. Anytime-Valid Test via Confidence Sequences

The *weak* null hypothesis can be tested in a closely related manner using confidence sequences (CS). Let $(L_t, U_t)_{t \geq s^v + 1}$ be a confidence sequence for the running average $\bar{\delta}_t^{v,c} := \frac{1}{t-s^v}\sum_{u=s^v+1}^{t} \delta_u$, that is,

$$\mathbb{P}\big(\forall t \geq s^v + 1 : L_t \leq \bar{\delta}_t^{v,c} \leq U_t\big) \geq 1 - \alpha.$$

Then the stopping time

$$\tau_{\text{w}}^{v,c} := \inf\{t \geq s^v + 1 : L_t > 0\}$$

defines an anytime-valid test of the weak null hypothesis, with guarantees analogous to those of the betting-based test for the strong null.

In practice, we use the empirical Bernstein CS (Howard et al., 2021), which has recently been successfully applied to online model comparison (Schirmer et al., 2025) and sequential forecasters evaluation (Choe & Ramdas, 2024).

*Remark* 3.1. Although betting-based tests and confidence-sequence-based tests appear different here, they are closely connected. Confidence sequences are typically constructed from nonnegative (super)martingales via Ville's inequality, and conversely, betting processes can be inverted to yield confidence sequences. See Orabona & Jun (2023) for constructions of confidence sequences from betting strategies similar to ours, and (Howard et al., 2021; Choe & Ramdas, 2024; Ramdas & Wang, 2025) for a general treatment of confidence sequences derived from supermartingales.

*Remark* 3.2. To test for a strictly positive advantage $\varepsilon > 0$, both approaches admit straightforward modifications. For the betting-based (strong) test, we use the $\varepsilon$-shifted wealth process $W_{\varepsilon,t}^{v,c} := \prod_{u=s^v+1}^{t} \left(1 + \beta_u(\Delta_u^{v,c} - \varepsilon)\right)$, with betting fractions constrained to $\beta_u \in [0, 1/(1+\varepsilon)]$. For the CS-based (weak) test, the stopping rule is modified to $L_t > \varepsilon$.

Algorithmic details are given in Algorithm 1. Details on UP approximation, the empirical Bernstein CS, the $\alpha_{v,c}$ schedule, and challenger generation are given in Appendix B.

---

**Algorithm 1** Anytime-Valid Online Decision Tree (AVT)

---

**Require:** Global level $\alpha$, minimum samples $n_{\min}$, advantage $\varepsilon \geq 0$, test type TEST $\in \{\text{BETTING}, \text{CS}\}$
1: Initialize root $v_0$; active leaves $\mathcal{L} \leftarrow \{v_0\}$
2: Set levels $\{\alpha^{v,c}\}$ with $\sum_{v,c} \alpha^{v,c} \leq \alpha$
3: $(S^{v,c}, \tau^{v,c}) \leftarrow \begin{cases} (W_\varepsilon^{v,c} \leftarrow 1, \ 1/\alpha^{v,c}), & \text{BETTING}, \\ (L^{v,c} \leftarrow -\infty, \ \varepsilon), & \text{CS} \end{cases}$
       ▷ Initialize e-process / CS and stopping threshold
4: **for** $t = 1, 2, \dots$ **do**
5:     Receive $(X_t, Y_t)$; route $X_t$ to leaf $v \in \mathcal{L}$
6:     $\hat{y}^v \leftarrow m_{t-1}^v(X_t)$       ▷ Incumbent prediction
7:     **for** each candidate split $c \in \mathcal{C}_v$ **do**
8:       $\hat{y}^{v_c} \leftarrow m_{t-1}^{v_c}(X_t)$     ▷ Challenger prediction
9:       $\Delta_t^{v,c} \leftarrow \ell(\hat{y}^v, Y_t) - \ell(\hat{y}^{v_c}, Y_t)$ ▷ Loss improvement
10:      $S^{v,c} \leftarrow \text{TESTUPDATE}(S^{v,c}, \Delta_t^{v,c})$ ▷ Update wealth (betting) via UP or CS bound
11:      Update challenger statistics with $(X_t, Y_t)$
12:     **end for**
13:     Update incumbent statistics with $(X_t, Y_t)$
14:     **if** $n_t(v) \geq n_{\min}$ **and** $\exists c : S^{v,c} > \tau^{v,c}$ **then**
15:      $c^\star \leftarrow \arg\max_c S^{v,c}$    ▷ Most significant challenger
16:      Split $v$ using $c^\star$; create children $v_L^{c^\star}, v_R^c$
17:      $\mathcal{L} \leftarrow (\mathcal{L} \setminus \{v\}) \cup \{v_L^{c^\star}, v_R^{c^\star}\}$     ▷ Commit split; replace leaf by its two children
18:     **end if**
19: **end for**

---

# 4. Theoretical Guarantees

We establish guarantees for the proposed sequential split-selection mechanism under two notions of improvement: a *strong* (anytime advantage) hypothesis $H_0^{v,c}$ and a *weak* (average advantage) hypothesis $H_{\text{w},0}^{v,c}$. These correspond respectively to testing-by-betting and confidence-sequence–based tests. Proofs and technical details are deferred to the appendix.

## 4.1. Anytime Validity and Global Error Control

Both tests are anytime-valid: they control Type I error under arbitrary, data-dependent stopping times and without independence assumptions.

**Theorem 4.1** (Anytime validity and global control). *Let* $\{\alpha^{v,c}\}_{v,c}$ *satisfy* $\sum_{v,c} \alpha^{v,c} \leq \alpha$. *Apply to each candidate split* $(v, c)$ *either (i) the betting-based test for* $H_0^{v,c}$ *or (ii) the confidence-sequence test for* $H_{\text{w},0}^{v,c}$, *with level* $\alpha^{v,c}$.

*Then, with probability at least* $1 - \alpha$, *no false split is ever committed relative to the tested global null hypothesis. This guarantee holds uniformly over time, and adaptive tree growth.*

## 4.2. Power: Finite Commitment Under Advantage

When a candidate split enjoys a persistent predictive advantage, both tests commit in finite time.

**Theorem 4.2** (Finite commitment under advantage). *Fix a candidate split* $(v, c)$. *Assume there exist* $\Delta > 0$ *and* $n_0 \geq 0$ *such that either the instantaneous advantage satisfies* $\delta_t^{v,c} \geq \Delta$ *for all* $t \geq s^v + n_0$, *or the average advantage satisfies* $\bar{\delta}_t^{v,c} \geq \Delta$ *for all* $t \geq s^v + n_0$. *Then, under the corresponding alternative, the stopping times* $\tau^{v,c}$ *and* $\tau_{\text{w}}^{v,c}$ *are finite almost surely, with rate* $\tilde{\mathcal{O}}(\log(1/\alpha^{v,c})/\Delta^2)$ *with high probability.*

## 4.3. Guarantees Under Stationary Data

Under stationary data, the proposed procedure enjoys a stronger guarantee: the expected predictive performance of the deployed tree improves monotonically over time.

**Theorem 4.3** (Strict monotonicity between and at commit times). *Assume* $(X, Y) \sim P$ *are i.i.d. and* $\ell(\hat{y}, y)$ *is convex in its first argument. Run Algorithm 1 and let* $0 = \tau_0 < \tau_1 < \tau_2 < \cdots$ *denote the (random) commit times, and write* $\hat{m}_t$ *for the deployed predictor.*

(Between commits.) *For all* $k \geq 0$ *and* $t \in (\tau_k, \tau_{k+1})$,

$$\mathbb{E}[\ell(\hat{m}_t(X), Y)] \leq \mathbb{E}[\ell(\hat{m}_{t-1}(X), Y)].$$

(At commits.) *There exists* $\varepsilon$ *such that if splits are committed only when the weak test certifies an advantage exceeding* $\varepsilon$, *then with probability at least* $1 - \alpha$, *for every commit time* $\tau_k$,

$$\mathbb{E}[\ell(\hat{m}_{\tau_k}(X), Y)] \leq \mathbb{E}\left[\ell(\hat{m}_{\tau_k^-}(X), Y)\right],$$

*where* $\hat{m}_{\tau_k^-}$ *denotes the pre-commit model.*

*Remark* 4.4. The monotone update property is independent of the testing procedure and follows solely from convexity and plug-in leaf updates; under strong convexity, these updates yield strict improvement. Similar monotonicity results for ensembles are established by Mattei & Garreau (2025). The role of anytime-valid testing in our method is to ensure that *structural* updates (committed splits) also preserve monotonicity under data-dependent stopping.

*Remark* 4.5. In contrast to the weak hypothesis, rejecting the strong hypothesis only certifies that a challenger outperforms the incumbent at some time, not necessarily at the (commit) stopping time. Ensuring monotonicity under the strong hypothesis therefore requires an additional *stability* condition, such as persistence of the advantage after it first appears. However, these conditions are required for the theoretical analysis; in practice, we observe monotonicity for both tests.

### 4.4. Warm-up experiment

We evaluate the practical implications of our stationarity guarantees on a controlled synthetic benchmark. Data are generated i.i.d. from a fixed, tree-structured distribution using the `RandomTree` generator introduced in the original Hoeffding Tree work (Domingos & Hulten, 2000). This generator constructs a random decision tree by recursively splitting features at random and assigning class labels at the leaves. We use 10 numerical and 10 categorical features and generate trees of depth 4, corresponding to at most 8 leaves.

We compare the HT with our Anytime-Valid Tree (AVT) based on betting tests ($AVT_B$) and confidence sequences ($AVT_{CS}$). Unless stated otherwise, we use the same default hyperparameters across all experiments, namely $n_{min} = 20$ and $\varepsilon = 0$. At each time step, generalisation performance is estimated using a fixed, independent holdout set of size 200,000 drawn from the same distribution.

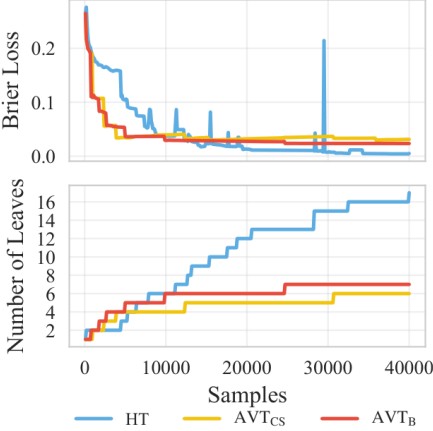

*Figure 1.* Generalisation error over time. The HT exhibits repeated performance drops, whereas anytime-valid methods remain stable.

**Results.** Figure 1 highlights a clear contrast between methods. The HT exhibits multiple abrupt drops in generalisation performance. Notably, this behaviour occurs despite the data being fully i.i.d. and stationary.

In contrast, both anytime-valid variants avoid such collapses and display stable, monotone improvement in generalisation, consistent with Theorem 4.3. The HT also commits substantially more splits than our methods, leading to larger and less stable trees. Our approach caps the number of leaves at 8, matching the size of the ground-truth tree.

Regarding the two anytime-valid methods, we observe that the variant based on the weak (average-based) hypothesis performs slightly worse in practice. This variant commits to splits later than the strong-hypothesis test, which translates into delayed adaptation and inferior predictive performance.

Finally, the HT's slight performance advantage at very large sample sizes is expected: By committing to many splits, it reduces approximation bias irrespective of split quality, and as estimation variance vanishes with more data, the model can perform well asymptotically.

## 5. Experiments

We evaluate our approach against (i) the Hoeffding Tree (HT) and (ii) its ensemble counterpart, Adaptive Random Forests (ARF). In both cases, we replace the HT base learner with our proposed Anytime-Valid Tree (AVT), yielding an Anytime-Valid Forest (AVF). In the main paper, we report results only for the betting-based variant ($AVT_B$), as it performs better overall than the CS version ($AVT_{CS}$), whose results are deferred to the Appendix C.1. Nevertheless, any positive advantage observed for $AVT_B$ over the baselines also carries over to $AVT_{CS}$. While our primary focus is improving HT and ARF, additional streaming baselines are reported in Appendix C.2. All methods are implemented using the `river` framework (Montiel et al., 2021).

We evaluate all methods on 12 data streams, comprising six regression and six classification tasks.

**Regression tasks:** `bike`, predicting daily bike-sharing usage; `chick`, modeling poultry growth trajectories; `fried`, forecasting food consumption patterns; `nzenergy`, forecasting regional energy consumption; `house`, predicting house prices; and `abalone`, estimating abalone age from physical measurements.

**Classification tasks:** `elec2`, predicting electricity price movements (up/down); `airlines`, predicting flight delays; `http-KDD99`, detecting HTTP network intrusions; `creditcard`, detecting fraudulent transactions; `rbfm100k`, a synthetic stream with gradual drift; and `hyper100k`, a synthetic stream with linear drift.

| | | | | | | Datasets | | | | | | |
|---|---|---|---|---|---|---|---|---|---|---|---|---|
| Method | rbfm100k | airlines | creditcard | hyper100k | elec2 | http | bike | chick | fried | nzenergy | house | abalone |
| | | | | | | Inference time (ms, mean ± std) | | | | | | |
| HT | 0.124±0.011 | 0.028±0.008 | 0.010±0.001 | 0.066±0.004 | 0.031±0.004 | 0.006±0.001 | 0.024±0.004 | 0.018±0.004 | 0.026±0.004 | 0.041±0.004 | 0.030±0.003 | 0.021±0.003 |
| $\mathrm{AVT_B}$ | 0.124±0.013 | 0.010±0.001 | 0.008±0.001 | 0.060±0.006 | 0.066±0.020 | 0.006±0.001 | 0.022±0.003 | 0.016±0.002 | 0.023±0.003 | 0.037±0.004 | 0.026±0.003 | 0.018±0.002 |
| ARF | 0.051±0.006 | 0.035±0.012 | 0.010±0.002 | 0.035±0.009 | 0.030±0.007 | 0.006±0.001 | 0.016±0.002 | 0.014±0.002 | 0.018±0.003 | 0.023±0.003 | 0.019±0.002 | 0.013±0.002 |
| $\mathrm{AVF_B}$ | 0.130±0.012 | 0.016±0.001 | 0.012±0.003 | 0.043±0.006 | 0.055±0.015 | 0.006±0.001 | 0.020±0.003 | 0.015±0.002 | 0.020±0.003 | 0.031±0.004 | 0.024±0.003 | 0.016±0.002 |
| | | | | | | Update time (ms, mean ± std) | | | | | | |
| HT | 0.074±0.009 | 0.078±0.026 | 0.110±0.023 | 0.147±4.795 | 0.094±0.334 | 0.017±0.001 | 0.078±0.023 | 0.064±0.006 | 0.046±0.018 | 0.608±0.670 | 0.139±0.075 | 0.056±0.010 |
| $\mathrm{AVT_B}$ | 5.22±0.71 | 9.61±3.16 | 0.132±0.038 | 2.48±4.09 | 18.66±55.84 | 0.020±0.003 | 3.16±0.29 | 1.20±0.23 | 4.78±0.26 | 25.82±28.79 | 5.52±0.48 | 1.89±0.22 |
| ARF | 0.228±2.449 | 2.18±14.21 | 0.079±0.012 | 0.469±1.107 | 0.379±0.870 | 0.028±0.006 | 0.013±0.005 | 0.010±0.002 | 0.028±0.007 | 0.074±0.079 | 0.021±0.004 | 0.011±0.001 |
| $\mathrm{AVF_B}$ | 7.65±0.40 | 10.65±4.40 | 0.208±0.052 | 3.82±4.42 | 27.94±82.30 | 0.034±0.010 | 0.401±0.034 | 0.190±0.033 | 0.486±0.028 | 3.06±3.63 | 0.533±0.035 | 0.277±0.023 |

*Table 1.* Per-example **inference and update time** (ms, mean ± std) across datasets.

For regression tasks, we enforce bounded losses via an adaptive scaling procedure. At each time step, we maintain an online estimate of the maximal observed loss and rescale the current loss, clipping it to $[0, 1]$. For both regression and classification, we use the squared loss $\ell(p, y) = \|p - y\|^2$.

**Evaluation protocol.** Evaluation is conducted using the standard prequential (test-then-train) protocol. We report classification accuracy and regression MAE, visualized in two ways: the left panels show the prequential performance curves averaged over 10 independent runs (with shaded 95% confidence intervals), together with the evolution of tree depth over time; for ensemble methods, we report the *average* tree depth across trees to enable direct comparison with single-tree models; and the right panels display the distribution of aggregate performance across runs.

**Model Performance and Complexity.** Figure 2 summarizes these results. Across the 12 data streams, our anytime-valid tree ($\mathrm{AVT_B}$) outperforms the standard HT on the majority of datasets, with abalone being the main exception where HT attains better aggregate performance. On several streams, including bike, fried, and hyper100k $\mathrm{AVT_B}$ is competitive with and occasionally surpasses the ensemble baseline ARF. Importantly, gains persist throughout the stream, as reflected by the prequential curves.

Among all approaches, $\mathrm{AVF_B}$ achieves the strongest overall performance across nearly all datasets, consistently outperforming the standard ARF baseline. Overall, performance differences become more pronounced on datasets exhibiting consistent drops or increased volatility over time, where betting-based methods adapt more effectively.

In terms of model complexity, $\mathrm{AVF_B}$ in addition of providing the best model also produces the smallest trees on average, even compared to single-tree methods. By contrast, $\mathrm{AVT_B}$ tends to grow deeper trees compared to its baseline HT. This is expected: in the absence of ensembling, a single tree must capture greater heterogeneity in the data stream, requiring additional depth to match the accuracy achievable by an ensemble.

**Computational Cost.** Table 1 reports per-instance inference and update times (mean ± std over 10 runs). Inference costs for $\mathrm{AVT_B}$ remain comparable to HT, since prediction involves only standard tree traversal. The primary computational overhead arises during updates, where betting wealth is maintained for each candidate split.

As expected, betting-based methods incur higher update costs due to (i) maintaining shadow challenger statistics, and (ii) estimating the Universal Portfolio wealth.

Despite this additional cost, runtimes remain within practical bounds. For most datasets, update times stay within a few milliseconds per instance. Even on more demanding streams (e.g., nzenergy), costs remain compatible with real-time processing.

Two factors further mitigate this overhead. First, $\mathrm{AVF_B}$ typically learns shallower trees, reducing the number of nodes requiring update; for example, it is even faster than its single-tree counterpart $\mathrm{AVT_B}$ on nzenergy. Second, computations across candidate splits and ensemble members are independent, making the method naturally amenable to parallelization. Overall, the results demonstrate a favorable computational trade-off for the observed performance gains.

## 6. Conclusion

We introduced an anytime-valid framework for split selection in online decision trees. Our approach corrects a fundamental limitation of existing methods based on fixed-sample concentration bounds by providing statistical guarantees that remain valid under data-dependent stopping rules, including non-stationary and dependent data streams. The resulting procedure serves as a drop-in replacement for standard Hoeffding Trees and integrates seamlessly into ensemble methods such as Adaptive Random Forests, a state-of-the-art bagging-based approach for data stream learning. Empirical evaluations on a diverse suite of real-world streaming datasets demonstrate that the resulting ensembles consistently outperform existing baselines while producing substantially smaller trees.

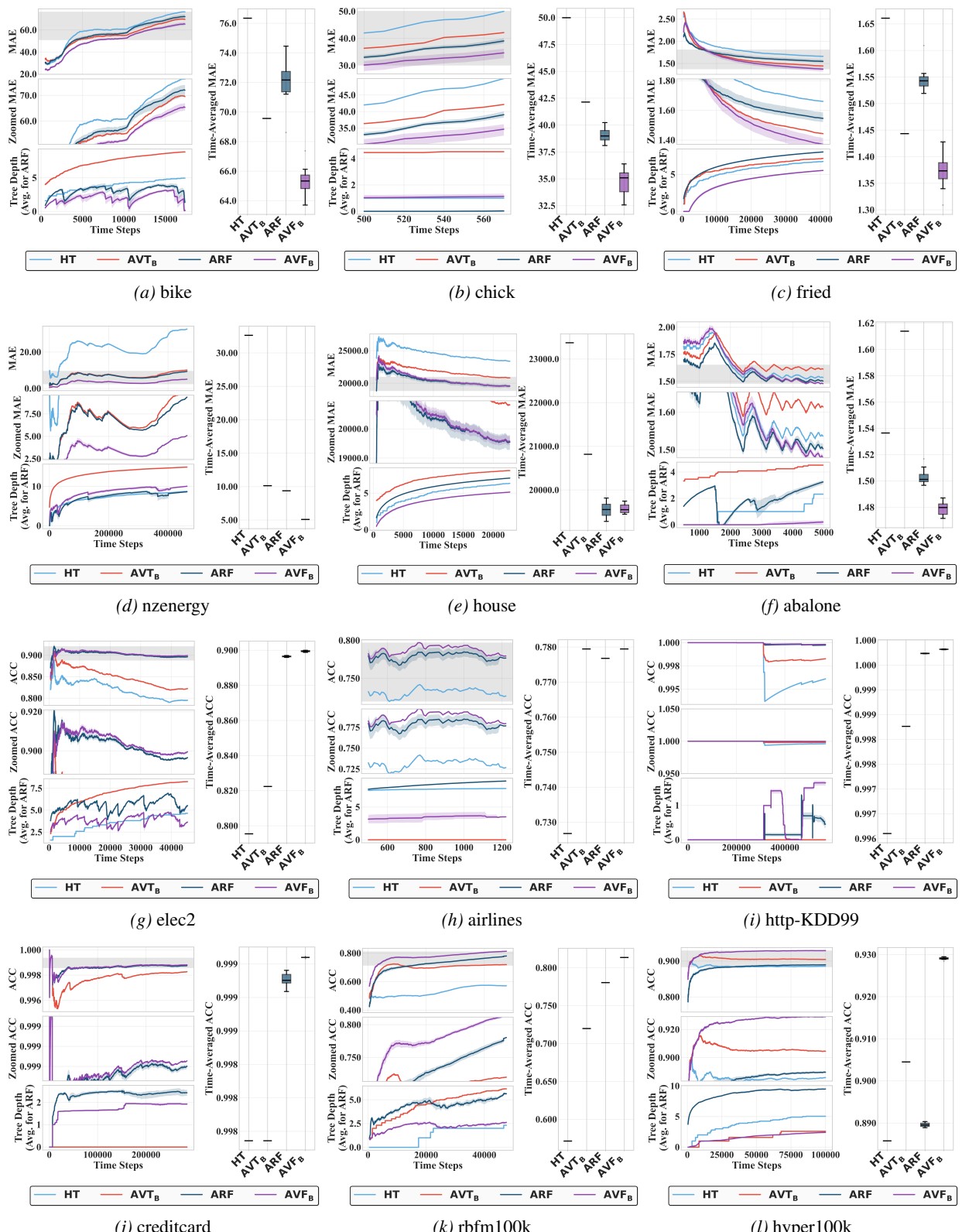

*Figure 2.* Regression datasets **(a)–(f)** and classification datasets **(g)–(l)**. Each panel shows prequential performance over 10 runs with $95\%$ confidence intervals, zoomed ambiguous regions, average tree depth, and aggregated metrics.

## Disclaimer

## Impact Statement

This paper presents work whose goal is to advance the field of Machine Learning. There are many potential societal consequences of our work, none which we feel must be specifically highlighted here.

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

# A. Proofs

## A.1. Basic measurability and boundedness

Fix a node–candidate pair $(v, c)$ and a test start time $s := s^{v,c}$. By construction, the incumbent and challenger predictions at time $t$ depend only on data observed up to time $t - 1$ and any algorithmic randomness revealed by then; thus they are $\mathcal{F}_{t-1}$-measurable. Since the loss $\ell$ is deterministic given a prediction and $Y_t$, the difference $\Delta_t^{v,c} = \ell(m_{t-1}^v, Y_t) - \ell(m_{t-1}^{v_c}, Y_t)$ is $\mathcal{F}_t$-measurable.

Assume $\Delta_t^{v,c} \in [-1, 1]$ a.s. and a predictable betting fraction $\beta_t^{v,c} \in [0, 1]$, i.e. $\beta_t^{v,c}$ is $\mathcal{F}_{t-1}$-measurable. Then the multiplicative factor satisfies $1 + \beta_t^{v,c}\Delta_t^{v,c} \in [0, 2]$ a.s., hence the wealth

$$W_{s-1}^{v,c} = 1, \qquad W_t^{v,c} = W_{t-1}^{v,c}\left(1 + \beta_t^{v,c}\Delta_t^{v,c}\right), \; t \geq s,$$

is well-defined and nonnegative.

## A.2. Proof of Lemma A.1

**Lemma A.1** (Anytime-valid single-split test). *Under $H_0^{v,c}$, the process $(W_t^{v,c})_{t \geq s^{v,c}-1}$ is a nonnegative $\mathcal{F}_t$-supermartingale. Consequently, for any $\alpha^{v,c} \in (0, 1)$,*

$$\mathbb{P}_{H_0^{v,c}}\left(\sup_{t \geq s^{v,c}} W_t^{v,c} \geq \frac{1}{\alpha^{v,c}}\right) \leq \alpha^{v,c}, \qquad \text{and hence} \qquad \mathbb{P}_{H_0^{v,c}}(\tau^{v,c} < \infty) \leq \alpha^{v,c}.$$

Under the null hypothesis $H_0^{v,c}$ we have $\mathbb{E}[\Delta_t^{v,c} \mid \mathcal{F}_{t-1}] \leq 0$ for all $t \geq s$. Using predictability of $\beta_t^{v,c}$ and $\mathcal{F}_{t-1}$-measurability of $W_{t-1}^{v,c}$,

$$\mathbb{E}[W_t^{v,c} \mid \mathcal{F}_{t-1}] = \mathbb{E}\left[W_{t-1}^{v,c}\left(1 + \beta_t^{v,c}\Delta_t^{v,c}\right) \,\Big|\, \mathcal{F}_{t-1}\right]$$
$$= W_{t-1}^{v,c}\left(1 + \beta_t^{v,c}\mathbb{E}[\Delta_t^{v,c} \mid \mathcal{F}_{t-1}]\right) \leq W_{t-1}^{v,c}.$$

Thus $(W_t^{v,c})_{t \geq s-1}$ is a nonnegative $\mathcal{F}_t$-supermartingale.

We now invoke Ville's inequality: if $(W_t)_{t \geq 0}$ is a nonnegative supermartingale with $W_0 = 1$, then for any $a > 0$, $\mathbb{P}(\sup_{t \geq 0} W_t \geq a) \leq 1/a$. Applying this with $a = 1/\alpha^{v,c}$ yields

$$\mathbb{P}_{H_0^{v,c}}\left(\sup_{t \geq s} W_t^{v,c} \geq \frac{1}{\alpha^{v,c}}\right) \leq \alpha^{v,c}.$$

Finally, $\{\tau^{v,c} < \infty\} \subseteq \{\sup_{t \geq s} W_t^{v,c} \geq 1/\alpha^{v,c}\}$, so $\mathbb{P}_{H_0^{v,c}}(\tau^{v,c} < \infty) \leq \alpha^{v,c}$ as claimed.

We can prove similarly the same result for the weak hypothesis $H_{\text{w},0}^{v,c}$. $\qquad \square$

## A.3. Countability under adaptive tree growth

We record a simple sufficient condition for countability, used by Theorem A.2. Suppose (i) at each time $t$ the algorithm maintains finitely many current nodes, and (ii) each node $v$ has a finite candidate class $C_v$. Then the set of all tests ever instantiated up to any finite time $t$ is finite; hence the set of tests over all times $t \in \mathbb{N}$ is a countable union of finite sets and therefore countable.

**Theorem A.2** (Global false-split control). *Assume that the set of all tests ever instantiated by the (possibly adaptive) tree-growing algorithm is almost surely countable, and that each level $\alpha^{v,c}$ is fixed at (or before) time $s^{v,c}$ (i.e. $\alpha^{v,c}$ is $\mathcal{F}_{s^{v,c}-1}$-measurable). If the allocation satisfies*

$$\sum_{(v,c)} \alpha^{v,c} \leq \alpha,$$

*then the probability of any false split over the entire lifetime of the tree is at most $\alpha$:*

$$\mathbb{P}(\exists\, (v, c) : H_0^{v,c} \text{ holds and } \tau^{v,c} < \infty) \leq \alpha.$$

## A.4. Proof of Theorem A.2

Let $\mathcal{T}$ denote the (a.s. countable) set of all tests instantiated by the algorithm, indexed by pairs $(v, c)$. For each test define the "false rejection" event

$$E^{v,c} := \{H_0^{v,c} \text{ holds and } \tau^{v,c} < \infty\}.$$

By Lemma A.1, $\mathbb{P}(E^{v,c}) \leq \alpha^{v,c}$ for each $(v, c) \in \mathcal{T}$, regardless of dependence across tests. Hence by the union bound over the countable family,

$$\mathbb{P}\left( \bigcup_{(v,c)\in\mathcal{T}} E^{v,c} \right) \leq \sum_{(v,c)\in\mathcal{T}} \mathbb{P}(E^{v,c}) \leq \sum_{(v,c)\in\mathcal{T}} \alpha^{v,c} \leq \alpha.$$

This proves the stated global control. $\qquad\square$

## A.5. Proof of Theorem 4.3

We prove monotone expected performance of the *deployed* predictor $(\hat{m}_t)_{t\geq 0}$ produced by the procedure under i.i.d. data and convex losses, on the event $\mathcal{E}$ that no false split is ever committed. For clarity, we focus on the classification setting; however, the result extends directly to any bounded-output prediction problem. Recall the theorem says:

**Theorem A.3** (Strict monotonicity between and at commit times). *Assume $(X, Y) \sim P$ are i.i.d. and $\ell(\hat{y}, y)$ is convex in its first argument. Run Algorithm 1 and let $0 = \tau_0 < \tau_1 < \tau_2 < \cdots$ denote the (random) commit times, and write $\hat{m}_t$ for the deployed predictor.*

(Between commits.) *For all $k \geq 0$ and $t \in (\tau_k, \tau_{k+1})$,*

$$\mathbb{E}[\ell(\hat{m}_t(X), Y)] \leq \mathbb{E}[\ell(\hat{m}_{t-1}(X), Y)].$$

(At commits.) *There exists $\varepsilon > 0$ such that if splits are committed only when the weak test certifies an advantage exceeding $\varepsilon$, then with probability at least $1 - \alpha$, for every commit time $\tau_k$,*

$$\mathbb{E}[\ell(\hat{m}_{\tau_k}(X), Y)] \leq \mathbb{E}\left[\ell(\hat{m}_{\tau_k^-}(X), Y)\right]$$

*where $\hat{m}_{\tau_k^-}$ denotes the pre-commit model.*

**Standing notation.** Let $(\mathcal{F}_t)_{t\geq 0}$ be the natural filtration generated by the stream and the internal randomness of the algorithm. At each time $t \geq 0$, $\hat{m}_t$ denotes the deployed tree predictor after processing $t$ samples; hence $\hat{m}_t$ is $\mathcal{F}_t$-measurable. Let $(X, Y) \sim P$ be an independent fresh draw, independent of $\mathcal{F}_t$. Define the (conditional) one-step-ahead risk

$$R_t := \mathbb{E}[\ell(\hat{m}_t(X), Y) \,|\, \mathcal{F}_t], \qquad \bar{R}_t := \mathbb{E}[\ell(\hat{m}_t(X), Y) \,|\, \mathcal{E}].$$

Note that $R_t$ is a random variable (measurable w.r.t. $\mathcal{F}_t$), while $\bar{R}_t$ is a deterministic scalar.

We decompose each update $t \mapsto t + 1$ into two parts: (i) a *parameter update* of the leaf statistics (empirical probabilities), and (ii) possibly a *structural update* (a committed split). We show that each part is non-increasing in expected risk on $\mathcal{E}$, which implies the claimed monotonicity.

### A.5.1. STEP 1: MARSHALL–PROSCHAN IMPLIES MONOTONE PLUG-IN RISK

The next lemma is the key tool controlling the fact that for convex losses, the expected risk of an empirical-mean predictor is nonincreasing with sample size.

**Lemma A.4** (Marshall–Proschan (Marshall & Proschan, 1964; Mattei & Garreau, 2025) monotonicity for empirical means). *Let $Z_1, \ldots, Z_n$ be exchangeable random variables in $[0, 1]$ and let $L : [0, 1] \to \mathbb{R}$ be convex. Define $\bar{Z}_n := \frac{1}{n}\sum_{i=1}^n Z_i$. Then for all $n \geq 2$,*

$$\mathbb{E}[L(\bar{Z}_n)] \leq \mathbb{E}[L(\bar{Z}_{n-1})].$$

*Proof.* Let $\mathbf{1} \in \mathbb{R}^n$ denote the all-ones vector and define

$$U = \begin{pmatrix} 0 & 1 & 1 & \cdots & 1 \\ 1 & 0 & 1 & \cdots & 1 \\ 1 & 1 & 0 & \cdots & 1 \\ \vdots & \vdots & \vdots & \ddots & \vdots \\ 1 & 1 & 1 & \cdots & 0 \end{pmatrix} \in \mathbb{R}^{n\times n}.$$

Then $U\mathbf{1} = (n - 1)\mathbf{1}$, hence $\mathbf{1} = \frac{1}{n-1}U\mathbf{1}$.

Let $Z = (Z_1, \ldots, Z_n)^\top$. Note that

$$\bar{Z}_n = \frac{1}{n}\mathbf{1}^\top Z.$$

Moreover, the $j$-th component of $U^\top Z$ equals $\sum_{i\neq j} Z_i$, hence the vector of leave-one-out empirical means is $\frac{1}{n-1}U^\top Z$, and

$$\frac{1}{n}\mathbf{1}^\top\left(\frac{1}{n-1}U^\top Z\right) = \frac{1}{n(n-1)}(U\mathbf{1})^\top Z = \frac{1}{n}\mathbf{1}^\top Z = \bar{Z}_n.$$

Therefore

$$\bar{Z}_n = \frac{1}{n} \sum_{j=1}^{n} \bar{Z}_{n-1}^{(-j)}, \qquad \bar{Z}_{n-1}^{(-j)} := \frac{1}{n-1} \sum_{i \neq j} Z_i,$$

i.e., $\bar{Z}_n$ is a uniform convex combination of the leave-one-out means.

Since $L$ is convex, Jensen's inequality gives

$$L(\bar{Z}_n) \leq \frac{1}{n} \sum_{j=1}^{n} L\left( \bar{Z}_{n-1}^{(-j)} \right).$$

Taking expectations and using exchangeability (the law of $\bar{Z}_{n-1}^{(-j)}$ does not depend on $j$) yields

$$\mathbb{E}[L(\bar{Z}_n)] \leq \frac{1}{n} \sum_{j=1}^{n} \mathbb{E}\left[ L(\bar{Z}_{n-1}^{(-j)}) \right] = \mathbb{E}\left[ L(\bar{Z}_{n-1}^{(-1)}) \right] = \mathbb{E}\left[ L(\bar{Z}_{n-1}) \right],$$

where the last equality uses $(Z_1, \ldots, Z_n) \overset{d}{=} (Z_2, \ldots, Z_n)$ under exchangeability. $\qquad \square$

**Lemma A.5** (Plug-in risk is nonincreasing in sample size). *Assume $Y_1, Y_2, \ldots$ are i.i.d. Bernoulli$(p)$. Let $\hat{p}_n := \frac{1}{n} \sum_{i=1}^{n} Y_i$. Assume that for each $y \in \{0, 1\}$, the map $\hat{y} \mapsto \ell(\hat{y}, y)$ is convex. Let $Y \sim$ Bernoulli$(p)$ be an independent fresh draw, independent of $(Y_i)_{i \geq 1}$. Then for all $n \geq 1$,*

$$\mathbb{E}[\ell(\hat{p}_{n+1}, Y)] \leq \mathbb{E}[\ell(\hat{p}_n, Y)].$$

*Proof.* Fix $y \in \{0, 1\}$ and define the convex function $L_y(\hat{y}) := \ell(\hat{y}, y)$. Apply Lemma A.4 to the exchangeable variables $Y_1, \ldots, Y_{n+1}$ with $L = L_y$:

$$\mathbb{E}\left[ L_y \left( \frac{1}{n+1} \sum_{i=1}^{n+1} Y_i \right) \right] \leq \mathbb{E}\left[ L_y \left( \frac{1}{n} \sum_{i=1}^{n} Y_i \right) \right].$$

Since $\hat{p}_{n+1} = \frac{1}{n+1} \sum_{i=1}^{n+1} Y_i$ and $\hat{p}_n = \frac{1}{n} \sum_{i=1}^{n} Y_i$, this becomes $\mathbb{E}[\ell(\hat{p}_{n+1}, y)] \leq \mathbb{E}[\ell(\hat{p}_n, y)]$. Now average over the independent $Y \sim$ Bernoulli$(p)$ to conclude

$$\mathbb{E}[\ell(\hat{p}_{n+1}, Y)] \leq \mathbb{E}[\ell(\hat{p}_n, Y)].$$

$\qquad \square$

### A.5.2. STEP 2: PARAMETER UPDATES (NO SPLIT) CANNOT INCREASE EXPECTED RISK

We now lift Lemma A.5 from a single Bernoulli mean to the full tree. When no split is committed, each new observation updates the statistics of exactly one leaf, and convexity ensures that this update cannot increase the expected risk.

**Lemma A.6** (Leaf-statistics update is risk-nonincreasing). *Let $(X, Y) \sim P$ be a fresh draw, independent of the data stream. Fix a time $t \geq 1$ such that no structural change (split commit) occurs between times $t - 1$ and $t$; only leaf statistics are updated using the observation $(X_t, Y_t)$. Then*

$$\mathbb{E}[\ell(\hat{m}_t(X), Y)] \leq \mathbb{E}[\ell(\hat{m}_{t-1}(X), Y)].$$

*Proof.* Fix such a time $t$ and let $\mathcal{L} := \mathcal{L}_{t-1}$ be the (random) set of leaves of the deployed tree at time $t - 1$, with associated regions $\{A(u) : u \in \mathcal{L}\}$. Since there is no structural update between $t - 1$ and $t$, the deployed partition $\{A(u)\}_{u \in \mathcal{L}}$ is the same at both times.

**A sigma-field that fixes the deployed partition.** For each leaf $u \in \mathcal{L}$, let $\tau(u) \leq t - 1$ denote its (random) creation time, and define the post-creation index set and corresponding sample count

$$I_{t-1}(u) := \{r : \tau(u) < r \leq t - 1, \ X_r \in A(u)\}, \qquad N_{t-1}(u) := |I_{t-1}(u)|.$$

We define the sigma-field $\mathcal{H}$ generated by the tree structure and split thresholds up to the creation of the leaf $u$. By construction, conditional on $\mathcal{H}$ the deployed partition $\{A(u) : u \in \mathcal{L}\}$ and all associated routing information are fixed and deterministic. Moreover, for each leaf $u$, the empirical statistic $\hat{p}_{t-1}(u)$ depends only on labels $(Y_r)_{r \in I_{t-1}(u)}$ observed *after* the creation of $u$. In particular, these labels are independent of the samples used to determine the partition and to perform the split-selection tests.

**Leafwise decomposition of the fresh risk.** Because $\hat{m}_{t-1}$ is piecewise-constant on the deployed partition, conditioning on $\mathcal{H}$ yields

$$\mathbb{E}[\ell(\hat{m}_{t-1}(X), Y) \,|\, \mathcal{H}] = \sum_{u \in \mathcal{L}} \mathbb{P}(X \in A(u) \,|\, \mathcal{H}) \ \mathbb{E}[\ell(\hat{p}_{t-1}(u), Y) \,|\, \mathcal{H}, \ X \in A(u)].$$

The analogous decomposition holds at time $t$ with $\hat{p}_t(u)$ in place of $\hat{p}_{t-1}(u)$.

**Within each leaf, post-creation labels are i.i.d. given $\mathcal{H}$.** By the honesty/no-reuse property mentioned above, which stated that the empirical statistics depend only on labels observed after the creation of the nodes, for every $s \geq \tau(u) + 1$,

$$\hat{p}_s(u) = \frac{1}{N_s(u)} \sum_{r \in I_s(u)} Y_r, \qquad I_s(u) := \{r : \tau(u) < r \leq s, \ X_r \in A(u)\}.$$

Fix $u \in \mathcal{L}$. Conditional on $\mathcal{H}$ and on $\{X \in A(u)\}$, the set $A(u)$ and the time $\tau(u)$ are fixed, and the stream is i.i.d. Hence the labels $(Y_r)_{r \in I_s(u)}$ are i.i.d. under $\mathbb{P}(\,\cdot\,|\,\mathcal{H}, \ X \in A(u))$ with parameter

$$p(u) := \mathbb{P}(Y = 1 \,|\, X \in A(u), \ \mathcal{H}).$$

**Apply Marshall–Proschan via Lemma A.5.** Fix $u \in \mathcal{L}$ and condition further on $\{N_{t-1}(u) = n\}$, which is $\mathcal{H}$-measurable. If $X_t \notin A(u)$ then leaf $u$ is not updated and $\hat{p}_t(u) = \hat{p}_{t-1}(u)$. If $X_t \in A(u)$ then $N_t(u) = n + 1$ and $\hat{p}_t(u)$ is the empirical mean of the $n + 1$ i.i.d. Bernoulli$(p(u))$ labels routed to $u$. Therefore, by Lemma A.5,

$$\mathbb{E}[\ell(\hat{p}_t(u), Y) \,|\, \mathcal{H}, \ X \in A(u), \ N_{t-1}(u) = n, \ X_t \in A(u)] \leq \mathbb{E}[\ell(\hat{p}_{t-1}(u), Y) \,|\, \mathcal{H}, \ X \in A(u), \ N_{t-1}(u) = n],$$

while equality holds on $\{X_t \notin A(u)\}$. Averaging over the event $\{X_t \in A(u)\}$ and then over $N_{t-1}(u)$ yields the leafwise inequality

$$\mathbb{E}[\ell(\hat{p}_t(u), Y) \,|\, \mathcal{H}, \ X \in A(u)] \leq \mathbb{E}[\ell(\hat{p}_{t-1}(u), Y) \,|\, \mathcal{H}, \ X \in A(u)].$$

**Conclude by mixing over leaves and removing conditioning.** Plugging the leafwise inequality into the leaf-mixture decomposition gives

$$\mathbb{E}[\ell(\hat{m}_t(X), Y) \,|\, \mathcal{H}] \leq \mathbb{E}[\ell(\hat{m}_{t-1}(X), Y) \,|\, \mathcal{H}].$$

Finally, take expectations over $\mathcal{H}$ (tower property) to conclude

$$\mathbb{E}[\ell(\hat{m}_t(X), Y)] \leq \mathbb{E}[\ell(\hat{m}_{t-1}(X), Y)],$$

as claimed. $\qquad\square$

### A.5.3. STEP 3: STRUCTURAL UPDATES (COMMITTED SPLITS) REDUCE EXPECTED RISK ON $\mathcal{E}$

We now handle the remaining case: the algorithm commits a split at time $t$. Although the theorem statement does not explicitly mention incumbents and challengers, the commit rule is defined via an anytime-valid test comparing the prequential performance of the currently deployed leaf predictor to that of its challenger split. We use this comparison only inside the proof to show that the commit step is risk-improving on $\mathcal{E}$.

**Prequential comparison and instantaneous advantage.** Fix a candidate split $(v, c)$ that is defined at time $t$ (i.e., its incumbent leaf $v$ exists in the deployed tree at time $t - 1$ and the challenger children $v_c^L, v_c^R$ are well-defined). Both incumbent and challenger predictors are prequential and $\mathcal{F}_{t-1}$-measurable:

$$m_{t-1}^v(x) = p_{t-1}(v), \qquad m_{t-1}^{v_c}(x) = \begin{cases} p_{t-1}(v_c^L), & x \in R(v_c^L), \\ p_{t-1}(v_c^R), & x \in R(v_c^R). \end{cases}$$

After observing $(X_t, Y_t)$, define the bounded prequential loss difference

$$\Delta_t^{v,c} := \ell\big(m_{t-1}^v(X_t), Y_t\big) - \ell\big(m_{t-1}^{v_c}(X_t), Y_t\big) \in [-1, 1], \qquad \delta_t^{v,c} := \mathbb{E}[\Delta_t^{v,c} \mid \mathcal{F}_{t-1}].$$

**Prequential mean equals fresh-draw mean.** Let $(X, Y) \sim P$ be an independent fresh draw, independent of $\mathcal{F}_{t-1}$. The next lemma allows us to interpret $\delta_t^{v,c}$ as the conditional *fresh-draw* risk gap between the incumbent and challenger, which is the quantity needed to argue that committing the split improves risk.

**Lemma A.7** (Prequential conditional mean equals fresh-draw conditional mean). *Let $g : \mathcal{X} \times \mathcal{Y} \to \mathbb{R}$ be such that $g(x, y)$ is $\mathcal{F}_{t-1}$-measurable through its dependence on predictors at time $t - 1$. Then*

$$\mathbb{E}[g(X_t, Y_t) \mid \mathcal{F}_{t-1}] = \mathbb{E}[g(X, Y) \mid \mathcal{F}_{t-1}].$$

*Proof.* Conditional on $\mathcal{F}_{t-1}$, the pair $(X_t, Y_t)$ is independent of $\mathcal{F}_{t-1}$ and distributed as $P$, hence has the same conditional law as the independent fresh draw $(X, Y) \sim P$. Therefore the conditional expectations coincide. $\square$

Applying Lemma A.7 with $g(x, y) = \ell(m_{t-1}^v(x), y) - \ell(m_{t-1}^{v_c}(x), y)$ yields the identity

$$\delta_t^{v,c} = \mathbb{E}\big[\ell(m_{t-1}^v(X), Y) - \ell(m_{t-1}^{v_c}(X), Y) \,\big|\, \mathcal{F}_{t-1}\big],$$

i.e., $\delta_t^{v,c}$ is the conditional fresh-draw advantage of the challenger we use in the next sections, with $\Delta_t^{v,c} = \ell(m_{t-1}^v(X), Y) - \ell(m_{t-1}^{v_c}(X), Y)$

**A bounded-drift correction for the weak (average-advantage) test.** The weak test controls a running average of the advantage process $\delta_t^{v,c}$, whereas committing a split requires a guarantee on the *instantaneous* advantage at the commit time. We bridge this gap by showing that, between committed splits, the process $t \mapsto \delta_t^{v,c}$ admits a uniformly bounded one-step drift. This allows us to introduce a correction threshold $\epsilon$ such that a positive average advantage implies a positive instantaneous advantage at the commit time.

We present two ways of selecting $\epsilon$: a conservative worst-case bound and a fully adaptive, data-dependent alternative. We emphasize that this threshold is introduced solely for theoretical control. In practice, the effective drift is often much smaller, and empirical results suggest that the required correction can be negligible and in some cases even set to zero while still preserving monotonicity.

**Assumption A.8** (Minimum leaf sample size (weak-test stability)). We work with Brier loss $\ell(\hat{y}, y) = (\hat{y} - y)^2$ and leaf predictions and target output in $[0, 1]$. A candidate $(v, c)$ is eligible for testing or commitment only once all leaf statistics used by the incumbent and challenger predictors on the region affected by the split have received at least $n_{\min} \geq 1$ post-creation samples.

**Lemma A.9** (Bounded one-step drift of the prequential advantage (between commits)). *Assume Brier loss and Assumption A.8. Fix a candidate $(v, c)$ and a time $t$ such that* no structural update (commit) *occurs between times $t - 1$ and $t$. Then*

$$\left|\delta_{t+1}^{v,c} - \delta_t^{v,c}\right| \leq \rho, \qquad where \qquad \rho := \frac{4}{n_{\min} + 1},$$

*almost surely.*

*Proof.* For each realized history, define the fresh-draw loss-gap functions

$$h_{t-1}(x, y) := \ell(m_{t-1}^v(x), y) - \ell(m_{t-1}^{v_c}(x), y), \qquad h_t(x, y) := \ell(m_t^v(x), y) - \ell(m_t^{v_c}(x), y).$$

Under the i.i.d. assumption and with $(X, Y) \sim P$ independent of the stream,

$$\delta_t^{v,c} = \mathbb{E}[h_{t-1}(X, Y) \mid \mathcal{F}_{t-1}], \qquad \delta_{t+1}^{v,c} = \mathbb{E}[h_t(X, Y) \mid \mathcal{F}_t].$$

Since the distribution of the fresh draw is fixed, the same identities may be viewed as integrals of the random functions $h_{t-1}$ and $h_t$ against $P$. Hence

$$|\delta_{t+1}^{v,c} - \delta_t^{v,c}| \leq \mathbb{E}[|h_t(X, Y) - h_{t-1}(X, Y)| \mid \mathcal{F}_t] \leq \|h_t - h_{t-1}\|_\infty.$$

It remains to bound the uniform change in the loss-gap function.

Because no structural update affects the comparison, only empirical-mean leaf statistics are updated. If a leaf mean $\hat{p}_n \in [0, 1]$ with $n \geq n_{\min}$ receives one additional label $Y_t \in [0, 1]$, then

$$|\hat{p}_{n+1} - \hat{p}_n| = \frac{|Y_t - \hat{p}_n|}{n + 1} \leq \frac{1}{n_{\min} + 1}.$$

For Brier loss on $[0, 1]$,

$$|(\hat{y} - y)^2 - (\hat{y}' - y)^2| = |\hat{y} - \hat{y}'|\,|\hat{y} + \hat{y}' - 2y| \leq 2|\hat{y} - \hat{y}'|.$$

Thus the loss of the incumbent changes by at most $2/(n_{\min} + 1)$ uniformly in $(x, y)$, and the loss of the challenger changes by at most the same amount. By the triangle inequality,

$$\|h_t - h_{t-1}\|_\infty \leq \frac{4}{n_{\min} + 1}.$$

Combining the displays proves the claim. $\qquad\square$

**From average advantage to instantaneous advantage.** Fix a candidate $(v, c)$ and suppose it is compared over a time interval during which no structural update affecting the relevant routing occurs. Let

$$\bar{\Delta}_t^{v,c} := \frac{1}{t - s^{v,c}} \sum_{r=s^{v,c}+1}^{t} \Delta_r^{v,c}, \qquad \bar{\delta}_t^{v,c} := \frac{1}{t - s^{v,c}} \sum_{r=s^{v,c}+1}^{t} \delta_r^{v,c}.$$

The next lemma converts a lower bound on the running average into a lower bound on the final instantaneous advantage.

**Lemma A.10** (Average-to-last transfer under bounded drift). *Fix a candidate $(v, c)$ and a time $t \geq s^{v,c} + 1$. Assume that for all $r \in \{s^{v,c} + 1, \ldots, t - 1\}$, $|\delta_{r+1}^{v,c} - \delta_r^{v,c}| \leq \rho$. Then*

$$\delta_t^{v,c} \geq \bar{\delta}_t^{v,c} - \rho\,\frac{t - s^{v,c} - 1}{2}.$$

*In particular, if $\bar{\delta}_t^{v,c} \geq \rho(t - s^{v,c} - 1)/2 + \varepsilon$, then $\delta_t^{v,c} \geq \varepsilon$.*

*Proof.* Let $T = t - s^{v,c}$ and write $\delta_r = \delta_r^{v,c}$. For $k = 0, \ldots, T - 1$, the drift condition gives

$$\delta_{t-k} \leq \delta_t + k\rho, \qquad \text{or equivalently} \qquad \delta_t \geq \delta_{t-k} - k\rho.$$

Averaging over $k = 0, \ldots, T - 1$ yields

$$\delta_t \geq \frac{1}{T} \sum_{k=0}^{T-1} \delta_{t-k} - \frac{\rho}{T} \sum_{k=0}^{T-1} k = \bar{\delta}_t^{v,c} - \rho\frac{T - 1}{2}.$$

$\qquad\square$

**Corrected weak-test stopping rule.** Let $(L_t, U_t)$ be a confidence sequence for the running average $\bar{\delta}_t^{v,c}$, so that on its coverage event $L_t \leq \bar{\delta}_t^{v,c}$ for all $t \geq s^{v,c} + 1$. Combining Lemmas A.9–A.10, on the coverage event we have

$$\delta_t^{v,c} \geq L_t - \frac{\rho(t - s^{v,c} - 1)}{2}.$$

Therefore, to ensure an instantaneous margin $\varepsilon \geq 0$ at the commit time, one may use the corrected weak-test stopping time

$$\tau_{\text{w,bd}}^{v,c} := \inf\left\{ t \geq s^{v,c} + 1 : L_t > \varepsilon + \frac{\rho(t - s^{v,c} - 1)}{2} \right\}, \qquad \rho := \frac{4}{n_{\min} + 1}.$$

### A.5.4. ADAPTIVE BOUNDED-DRIFT CORRECTION USING LEAF COUNTS

The minimum leaf sample size requirement in Assumption A.8 can be removed by controlling the one-step variation of the instantaneous prequential advantage using the actual leaf counts involved in the shadow incumbent and shadow challenger predictions.

**Lemma A.11** (Adaptive one-step drift via leaf counts). *Assume i.i.d. data and Brier loss $\ell(\hat{y}, y) = (\hat{y} - y)^2$ with predictions in $[0, 1]$. Fix a candidate $(v, c)$ and a time $t \geq s^{v,c} + 1$ such that no structural update affects this comparison between $t - 1$ and $t$. Let $n_t^{\mathrm{inc}}$ and $n_t^{\mathrm{chal}}$ denote the numbers of samples in the shadow incumbent leaf and the relevant shadow challenger leaf immediately before their update at time $t$. Then*

$$\left| \delta_{t+1}^{v,c} - \delta_t^{v,c} \right| \leq \rho_t := \frac{2}{n_t^{\mathrm{inc}} + 1} + \frac{2}{n_t^{\mathrm{chal}} + 1}.$$

*Proof.* Use the fresh-draw functions $h_{t-1}$ and $h_t$ from the proof of Lemma A.9. The empirical mean of the affected incumbent leaf changes by at most $1/(n_t^{\mathrm{inc}} + 1)$, and the empirical mean of the affected challenger leaf changes by at most $1/(n_t^{\mathrm{chal}} + 1)$. The Brier loss is 2-Lipschitz in its prediction argument on $[0, 1]$. Therefore

$$\| h_t - h_{t-1} \|_\infty \leq \frac{2}{n_t^{\mathrm{inc}} + 1} + \frac{2}{n_t^{\mathrm{chal}} + 1}.$$

As before, $|\delta_{t+1}^{v,c} - \delta_t^{v,c}| \leq \| h_t - h_{t-1} \|_\infty$, proving the claim. □

**Lemma A.12** (Average-to-last transfer under adaptive drift). *Let $(a_i)_{i=1}^n$ be a real sequence such that $|a_{i+1} - a_i| \leq \rho_i$ for $i = 1, \ldots, n - 1$. Then*

$$a_n \geq \frac{1}{n} \sum_{i=1}^n a_i - \frac{1}{n} \sum_{j=1}^{n-1} j \rho_j.$$

*Consequently, with $a_i = \delta_{s^{v,c}+i}^{v,c}$ and $n = t - s^{v,c}$,*

$$\delta_t^{v,c} \geq \bar{\delta}_t^{v,c} - \frac{1}{t - s^{v,c}} \sum_{j=1}^{t-s^{v,c}-1} j \, \rho_{s^{v,c}+j}.$$

*Proof.* For $i = 1, \ldots, n$, repeated use of the drift bounds gives

$$a_i \leq a_n + \sum_{j=i}^{n-1} \rho_j.$$

Averaging over $i$ yields

$$\frac{1}{n} \sum_{i=1}^n a_i \leq a_n + \frac{1}{n} \sum_{i=1}^n \sum_{j=i}^{n-1} \rho_j.$$

For a fixed $j$, the term $\rho_j$ appears exactly for $i = 1, \ldots, j$, hence exactly $j$ times. Thus

$$\frac{1}{n} \sum_{i=1}^n a_i \leq a_n + \frac{1}{n} \sum_{j=1}^{n-1} j \rho_j,$$

which rearranges to the desired inequality. □

**Adaptive weak-test stopping rule.** Combining Lemmas A.11 and A.12, on the CS coverage event it is enough to use

$$\tau_{\mathrm{w,ad}}^{v,c} := \inf \left\{ t \geq s^{v,c} + 1 : L_t > \varepsilon + \frac{1}{t - s^{v,c}} \sum_{j=1}^{t-s^{v,c}-1} j \, \rho_{s^{v,c}+j} \right\},$$

where

$$\rho_u := \frac{2}{n_u^{\mathrm{inc}} + 1} + \frac{2}{n_u^{\mathrm{chal}} + 1}.$$

This corrects the adaptive average-to-last penalty: the appropriate drift budget is the weighted sum $(t - s^{v,c})^{-1} \sum_j j \rho_{s^{v,c}+j}$, not the unweighted one-half sum.

**Stationary risk certificate as a direct alternative.** The weak-test corrections above are one route to certifying instantaneous commit-time advantage. Under stationarity, a cleaner alternative is to certify the current risk improvement directly.

**Lemma A.13** (Stationary risk certificate). *Assume $(X_t, Y_t)$ are i.i.d. with law $P$ and $Y \in \{1, \dots, K\}$. Fix a candidate split $(v, c)$ at a possible commit time $t$, and let $b_c(x) \in \{L, R\}$ denote the child selected by split c. Conditional on $X \in R(v)$, define*

$$\theta_{u,k}^{v,c} := \mathbb{P}(b_c(X) = u,\ Y = k \mid X \in R(v)), \qquad u \in \{L, R\},\ k \in [K].$$

*Let $q_{v,t-1}$ be the current shadow-incumbent prediction and let $q_{L,t-1}, q_{R,t-1}$ be the current shadow-challenger child predictions. Suppose $\Theta_t^{v,c}$ is a confidence set such that, on an event $\mathcal{C}_t^{v,c}$, $\theta^{v,c} \in \Theta_t^{v,c}$. Define*

$$\underline{G}_t^{v,c} := \inf_{\theta \in \Theta_t^{v,c}} \sum_{u \in \{L,R\}} \sum_{k=1}^{K} \theta_{u,k} \Big( \ell(q_{v,t-1}, k) - \ell(q_{u,t-1}, k) \Big).$$

*On $\mathcal{C}_t^{v,c}$, if $\underline{G}_t^{v,c} \geq 0$, then the split satisfies the instantaneous commit certificate*

$$\mathbb{E}\big[ \ell(m_{t-1}^v(X), Y) - \ell(m_{t-1}^{v_c}(X), Y) \mid \mathcal{F}_{t-1} \big] \geq 0.$$

*If $\underline{G}_t^{v,c} \geq \varepsilon > 0$, then the unconditional fresh-draw risk improvement is at least $\varepsilon\, \mathbb{P}(X \in R(v))$.*

*Proof.* The incumbent and challenger agree outside $R(v)$. Hence the unconditional fresh-draw risk gap equals

$$\mathbb{P}(X \in R(v)) \sum_{u \in \{L,R\}} \sum_{k=1}^{K} \theta_{u,k}^{v,c} \Big( \ell(q_{v,t-1}, k) - \ell(q_{u,t-1}, k) \Big).$$

On $\mathcal{C}_t^{v,c}$, the true vector $\theta^{v,c}$ belongs to $\Theta_t^{v,c}$, so the conditional-on-$R(v)$ gap is at least $\underline{G}_t^{v,c}$. If this lower bound is nonnegative, the unconditional gap is nonnegative. If the lower bound is at least $\varepsilon$, the unconditional gap is at least $\varepsilon\, \mathbb{P}(X \in R(v))$. $\qquad\square$

**Commit is risk-improving on $\mathcal{E}$.** We now isolate the structural effect of committing a split. Let $\hat{m}_t^{sp}$ be the intermediate deployed predictor obtained before the leaf-statistics update at time $t$ but after committing the split, and let $\hat{m}_t$ be the deployed predictor after committing the split.

**Lemma A.14** (Committing an advantageous split reduces conditional one-step risk on $\mathcal{E}$). *Fix a commit time $t$ at which the algorithm commits a split $(v, c)$. Assume that on the event $\mathcal{E}$, the committed split satisfies the instantaneous fresh-draw advantage certificate*

$$\delta_t^{v,c} := \mathbb{E}\big[ \ell\big(m_{t-1}^v(X), Y\big) - \ell\big(m_{t-1}^{v_c}(X), Y\big) \,\big|\, \mathcal{F}_{t-1} \big] \geq 0, \qquad\qquad (\star)$$

*where $(X, Y) \sim P$ is an independent fresh draw. Then, on $\mathcal{E}$,*

$$\mathbb{E}[\ell(\hat{m}_t^{sp}(X), Y) \mid \mathcal{F}_{t-1}] \leq \mathbb{E}[\ell(\hat{m}_{t-1}(X), Y) \mid \mathcal{F}_{t-1}] \quad a.s.$$

*Consequently,*

$$\mathbb{E}[\ell(\hat{m}_t^{sp}(X), Y)] \leq \mathbb{E}[\ell(\hat{m}_{t-1}(X), Y)].$$

*If $(\star)$ holds with margin $\varepsilon > 0$, then the unconditional structural risk decrease is at least $\varepsilon$ when $\varepsilon$ is an unconditional certificate, and at least $\varepsilon\, \mathbb{P}(X \in R(v))$ when $\varepsilon$ is a certificate conditional on $X \in R(v)$.*

*Proof.* The structural update replaces the incumbent prediction by the challenger prediction only on the affected region $R(v)$; outside $R(v)$, the two deployed predictors agree. Therefore,

$$\mathbb{E}[\ell(\hat{m}_{t-1}(X), Y) - \ell(\hat{m}_t^{sp}(X), Y) \mid \mathcal{F}_{t-1}]$$
$$= \mathbb{E}\big[ \ell\big(m_{t-1}^v(X), Y\big) - \ell\big(m_{t-1}^{v_c}(X), Y\big) \,\big|\, \mathcal{F}_{t-1} \big] = \delta_t^{v,c}.$$

The equality uses the fact that $m_{t-1}^v$ and $m_{t-1}^{v_c}$ agree outside $R(v)$ by construction, and that both predictors are $\mathcal{F}_{t-1}$-measurable. By $(\star)$, the right-hand side is nonnegative on $\mathcal{E}$, which proves the conditional risk inequality. Taking expectations gives the unconditional claim. The margin statements follow from the same display. If the certificate is stated conditional on $X \in R(v)$, the unconditional gap is the conditional gap multiplied by $\mathbb{P}(X \in R(v))$. $\qquad\square$

A.5.5. STEP 4: CONCLUDE THE THEOREM

*Proof of Theorem 4.3.* Fix $t \geq 1$. There are two cases.

**Case 1: no split is committed at time** $t$**.** Then $\hat{m}_t$ differs from $\hat{m}_{t-1}$ only through leaf-statistic updates. Lemma A.6 yields

$$\mathbb{E}[\ell(\hat{m}_t(X), Y)] \leq \mathbb{E}[\ell(\hat{m}_{t-1}(X), Y)].$$

**Case 2: a split is committed at time** $t$**.** Write $\hat{m}_t^{sp}$ for the intermediate model after structural commit but before the leaf-statistic updates. By Lemma A.14 on $\mathcal{E}$,

$$\mathbb{E}[\ell(\hat{m}_t^{sp}(X), Y)] \leq \mathbb{E}[\ell(\hat{m}_{t-1}(X), Y)],$$

and by Lemma A.6,

$$\mathbb{E}[\ell(\hat{m}_t(X), Y)] \leq \mathbb{E}[\ell(\hat{m}_t^{sp}(X), Y)].$$

Combining gives

$$\mathbb{E}[\ell(\hat{m}_t(X), Y)] \leq \mathbb{E}[\ell(\hat{m}_{t-1}(X), Y)].$$

Since $t$ was arbitrary, monotonicity holds for all $t \geq 1$ and the sequence $\big(\mathbb{E}[\ell(\hat{m}_t(X), Y)]\big)_{t \geq 0}$ is nonincreasing. $\square$

## A.6. Proof of Theorem 4.2

We prove the two commitment guarantees in Theorem 4.2:

1. Under the **strong alternative**, the **betting/wealth** stopping time $\tau^{v,c}$ is finite, and admits a high-probability bound of order $\tilde{\mathcal{O}}(\log(1/\alpha^{v,c})/\Delta^2)$.

2. Under the **weak alternative**, the **CS-crossing** stopping time $\tau_{\mathrm{w}}^{v,c} := \inf\{t \geq s^{v,c} : L_t > 0\}$ is finite with probability at least $1 - \alpha^{v,c}$, and admits the same rate when instantiated with the (empirical Bernstein) confidence sequence (Choe & Ramdas, 2024) used.

**Global notation.**   Fix a node–candidate pair $(v,c)$ and suppress superscripts $(v,c)$. Let the test start time be $s := s^{v,c}$. For brevity write
$$\Delta_t := \Delta_t^{v,c} \in [-1,1], \qquad \delta_t := \mathbb{E}[\Delta_t \mid \mathcal{F}_{t-1}], \qquad t \geq s.$$
We also set $n := t - s + 1$ for the *effective sample size* at time $t \geq s$.

The wealth of the betting-based test defined in (4) using the betting fraction (5) can be expressed as a continuous mixture over betting fractions,
$$W_t = \int_0^1 \prod_{i=s}^t \big(1 + \beta \, \Delta_i\big) \, d\pi(\beta), \tag{6}$$
where $\pi$ denotes the Jeffreys prior on $[0,1]$.

In practice, we have to resort to approximation. We use a finite discrete mixture that allows to maintain the theoretical guarantees.

### A.6.1. PART I: WEALTH (BETTING) STOPPING TIME UNDER THE STRONG ALTERNATIVE

We first prove the wealth-based guarantee for the actual discrete-mixture wealth used by the algorithm.

**Assumption A.15** (Persistent strong advantage for the original discrete mixture).   Fix $(v,c)$ and let $s = s^{v,c}$ be the first time at which this candidate's loss difference is used as test evidence. There exist $\Delta \in (0,1]$ and $n_0 \in \mathbb{N}$ such that, with $t_0 := s + n_0$,
$$\delta_t = \mathbb{E}[\Delta_t \mid \mathcal{F}_{t-1}] \geq \Delta \qquad \forall t \geq t_0.$$
Thus $n_0$ is the number of tested increments before the persistent advantage begins. The betting grid contains a point
$$\beta_\star \in \left[\frac{\Delta}{8}, \frac{\Delta}{4}\right]$$
with mixture weight $w_\star > 0$.

**Original discrete-mixture wealth and stopping rule.**   Let $\mathcal{B} = \{\beta_1, \ldots, \beta_K\} \subset [0,1)$ be the fixed grid of betting fractions and let $w_k > 0$ with $\sum_{k=1}^K w_k = 1$. The wealth is
$$W_t = \sum_{k=1}^K w_k \prod_{u=s}^t \big(1 + \beta_k \Delta_u\big), \qquad W_{s-1} = 1. \tag{7}$$

The betting stopping time is
$$\tau := \tau^{v,c} := \inf\left\{t \geq s : W_t \geq \frac{1}{\alpha^{v,c}}\right\}.$$

**Goal.**   We show that for every $\eta \in (0,1)$, with probability at least $1 - \eta$,
$$\tau \leq t_0 + N_\eta - 1,$$
where
$$N_\eta := \left\lceil \frac{32}{\Delta^2}\left(\log\frac{1}{\alpha^{v,c}} + \log\frac{1}{w_\star} + B_0 + \log\frac{1}{\eta}\right)\right\rceil, \qquad B_0 := n_0 \log\frac{1}{1-\beta_\star}. \tag{8}$$

STEP 0: LOWER-BOUND THE ORIGINAL MIXTURE BY ONE FAVORABLE GRID POINT

For every $t \geq s$, the mixture contains the component indexed by $\beta_\star$, so deterministically

$$W_t \geq w_\star \prod_{u=s}^{t} \left(1 + \beta_\star \Delta_u\right). \tag{9}$$

Because $\Delta_u \in [-1, 1]$ and $\beta_\star < 1$, each factor is strictly positive:

$$1 + \beta_\star \Delta_u \geq 1 - \beta_\star > 0.$$

Thus we may take logarithms. At time $t = t_0 + N - 1$, split the log-product into the pre-advantage block $u = s, \ldots, t_0 - 1$ and the post-advantage block $u = t_0, \ldots, t_0 + N - 1$:

$$\log W_{t_0 + N - 1} \geq \log w_\star + \sum_{u=s}^{t_0 - 1} \log(1 + \beta_\star \Delta_u) + \sum_{u=t_0}^{t_0 + N - 1} \log(1 + \beta_\star \Delta_u). \tag{10}$$

STEP 1: CONTROL THE STALE PRE-ADVANTAGE HISTORY

No sign or mean condition is imposed before $t_0$. Nevertheless, boundedness alone gives a deterministic lower bound. Since $\Delta_u \geq -1$,

$$1 + \beta_\star \Delta_u \geq 1 - \beta_\star,$$

and hence

$$\log(1 + \beta_\star \Delta_u) \geq \log(1 - \beta_\star) = -\log \frac{1}{1 - \beta_\star}.$$

There are $n_0 = t_0 - s$ pre-advantage tested increments, so

$$\sum_{u=s}^{t_0 - 1} \log(1 + \beta_\star \Delta_u) \geq -n_0 \log \frac{1}{1 - \beta_\star} = -B_0. \tag{11}$$

This is the only cost of using the original single-start wealth rather than a restart mixture. We could, in fact, circumvent this cost entirely by utilizing the restart mixture defined below:

For late-emerging advantages, one may allocate capital across restart times. For deterministic weights $\pi_r \geq 0$, $\sum_{r=s}^{\infty} \pi_r \leq 1$, define

$$\widetilde{W}_t = \sum_{r=s}^{t} \pi_r \sum_{k=1}^{K} w_k \prod_{u=r}^{t} (1 + \beta_k \Delta_u) + \sum_{r>t} \pi_r.$$

Each started component is an e-process under the strong null, and unstarted capital is held in cash, so $(\widetilde{W}_t)$ is a nonnegative supermartingale. If the advantage starts at $t_0$, the component $r = t_0$ pays a penalty $\log(1/\pi_{t_0})$ instead of the stale-history term. This is a valid optional variant but is a different algorithmic wealth process.

STEP 2: CONCENTRATION OF THE POST-ADVANTAGE CUMULATIVE LOSS DIFFERENCE

Fix a block length $N \geq 1$ and define, for $u \geq t_0$,

$$X_u := \Delta_u - \delta_u.$$

Then $(X_u, \mathcal{F}_u)$ is a martingale difference sequence:

$$\mathbb{E}[X_u \mid \mathcal{F}_{u-1}] = \mathbb{E}[\Delta_u - \delta_u \mid \mathcal{F}_{u-1}] = \delta_u - \delta_u = 0.$$

Moreover, since $\Delta_u \in [-1, 1]$, its conditional mean $\delta_u$ also lies in $[-1, 1]$, and $X_u$ lies conditionally in an interval of length at most 2. By the conditional Hoeffding lemma, for every $\lambda > 0$,

$$\mathbb{E}[\exp(-\lambda X_u) \mid \mathcal{F}_{u-1}] \leq \exp\left(\frac{\lambda^2}{2}\right).$$

Iterating this inequality over $u = t_0, \ldots, t_0 + N - 1$ gives

$$\mathbb{E}\left[\exp\left(-\lambda \sum_{u=t_0}^{t_0+N-1} X_u\right)\right] \leq \exp\left(\frac{N\lambda^2}{2}\right).$$

Therefore, by Chernoff's method, for any $a > 0$,

$$\mathbb{P}\left(\sum_{u=t_0}^{t_0+N-1} X_u \leq -a\right) \leq \exp\left(-\frac{a^2}{2N}\right).$$

Taking $a = \sqrt{2N \log(1/\eta)}$, we obtain an event $\mathcal{A}_N$ satisfying $\mathbb{P}(\mathcal{A}_N) \geq 1 - \eta$ on which

$$\sum_{u=t_0}^{t_0+N-1} X_u \geq -\sqrt{2N \log \frac{1}{\eta}}. \tag{12}$$

On $\mathcal{A}_N$, using $\delta_u \geq \Delta$ for $u \geq t_0$,

$$\sum_{u=t_0}^{t_0+N-1} \Delta_u = \sum_{u=t_0}^{t_0+N-1} \delta_u + \sum_{u=t_0}^{t_0+N-1} X_u$$

$$\geq N\Delta - \sqrt{2N \log \frac{1}{\eta}}. \tag{13}$$

If

$$N \geq \frac{8}{\Delta^2} \log \frac{1}{\eta}, \tag{14}$$

then

$$\sqrt{2N \log \frac{1}{\eta}} \leq \frac{N\Delta}{2},$$

and therefore on $\mathcal{A}_N$,

$$\sum_{u=t_0}^{t_0+N-1} \Delta_u \geq \frac{N\Delta}{2}. \tag{15}$$

STEP 3: CONVERT CUMULATIVE ADVANTAGE INTO LOG-WEALTH GROWTH

We use the elementary inequality

$$\log(1 + x) \geq x - x^2, \qquad x \in [-1/2, 1/2].$$

Indeed, for $g(x) = \log(1 + x) - x + x^2$, one has $g(0) = 0$ and $g'(x) = x(1 + 2x)/(1 + x)$, so $g$ is minimized at zero on $[-1/2, 1/2]$. Since $\beta_\star \leq \Delta/4 \leq 1/4$ and $\Delta_u \in [-1, 1]$, the product $x = \beta_\star \Delta_u$ lies in $[-1/4, 1/4]$. Hence

$$\log(1 + \beta_\star \Delta_u) \geq \beta_\star \Delta_u - \beta_\star^2 \Delta_u^2 \geq \beta_\star \Delta_u - \beta_\star^2.$$

Summing over the post-advantage block and using (15), on $\mathcal{A}_N$ we get

$$\sum_{u=t_0}^{t_0+N-1} \log(1 + \beta_\star \Delta_u) \geq \beta_\star \sum_{u=t_0}^{t_0+N-1} \Delta_u - N\beta_\star^2$$

$$\geq N\left(\frac{\beta_\star \Delta}{2} - \beta_\star^2\right). \tag{16}$$

Because $\beta_\star \leq \Delta/4$,

$$\beta_\star^2 \leq \frac{\beta_\star \Delta}{4},$$

so

$$\frac{\beta_\star \Delta}{2} - \beta_\star^2 \geq \frac{\beta_\star \Delta}{4}.$$

Because $\beta_\star \geq \Delta/8$,

$$\frac{\beta_\star \Delta}{4} \geq \frac{\Delta^2}{32}.$$

Therefore, on $\mathcal{A}_N$ and under (14),

$$\sum_{u=t_0}^{t_0+N-1} \log(1 + \beta_\star \Delta_u) \geq \frac{N\Delta^2}{32}. \tag{17}$$

STEP 4: SOLVE FOR THRESHOLD CROSSING

Combining (10), (11), and (17), on $\mathcal{A}_N$ we have

$$\log W_{t_0+N-1} \geq \log w_\star - B_0 + \frac{N\Delta^2}{32}. \tag{18}$$

Thus $W_{t_0+N-1} \geq 1/\alpha^{v,c}$ is guaranteed whenever

$$\log w_\star - B_0 + \frac{N\Delta^2}{32} \geq \log \frac{1}{\alpha^{v,c}},$$

or equivalently whenever

$$\frac{N\Delta^2}{32} \geq \log \frac{1}{\alpha^{v,c}} + \log \frac{1}{w_\star} + B_0. \tag{19}$$

The choice $N = N_\eta$ in (8) implies (19). It also implies the concentration requirement (14), because the bracket in (8) contains $\log(1/\eta)$ and $32 \geq 8$. Therefore,

$$\mathbb{P}\left(W_{t_0+N_\eta-1} \geq \frac{1}{\alpha^{v,c}}\right) \geq 1 - \eta.$$

By definition of $\tau$,

$$\mathbb{P}(\tau \leq t_0 + N_\eta - 1) \geq 1 - \eta.$$

**Almost sure finiteness.** For every $\eta \in (0,1)$, there is a deterministic finite $N_\eta$ such that

$$\mathbb{P}(\tau > t_0 + N_\eta - 1) \leq \eta.$$

Since $\{\tau = \infty\} \subseteq \{\tau > t_0 + N_\eta - 1\}$, we have $\mathbb{P}(\tau = \infty) \leq \eta$ for every $\eta \in (0,1)$. Letting $\eta \downarrow 0$ gives $\mathbb{P}(\tau = \infty) = 0$, so $\tau < \infty$ almost surely.

**Stale-history penalty.** Since $\beta_\star \leq \Delta/4 \leq 1/4$,

$$B_0 = n_0 \log \frac{1}{1-\beta_\star} \leq \frac{n_0 \beta_\star}{1-\beta_\star} \leq \frac{4}{3} n_0 \beta_\star \leq \frac{n_0 \Delta}{3}.$$

Thus the original single-start wealth pays an explicit stale-history penalty. Validity holds for any finite grid; the rate above additionally requires that the grid contain a betting fraction of order $\Delta$, namely $\beta_\star \in [\Delta/8, \Delta/4]$.

A.6.2. PART II: CS-CROSSING STOPPING TIME UNDER THE WEAK ALTERNATIVE

We now prove the CS part under the weak alternative.

**Running averages.** For $t \geq s$ define

$$\bar{\Delta}_t := \frac{1}{t-s+1} \sum_{u=s}^{t} \Delta_u, \qquad \bar{\delta}_t := \frac{1}{t-s+1} \sum_{u=s}^{t} \delta_u.$$

The weak null and weak alternative are:

$$H_{w,0} : \forall t \geq s, \ \bar{\delta}_t \leq 0, \qquad H_{w,1}(\Delta) : \exists t_0 \geq s \text{ s.t. } \forall t \geq t_0, \ \bar{\delta}_t \geq \Delta.$$

**Confidence sequence and stopping rule.** Let $(L_t, U_t)_{t \geq s}$ be a $(1 - \alpha^{v,c})$ CS for $\bar{\delta}_t$:

$$\mathbb{P}\Big(\forall t \geq s : \ L_t \leq \bar{\delta}_t \leq U_t\Big) \geq 1 - \alpha^{v,c}. \tag{20}$$

Define the CS-crossing stopping time

$$\tau_{\mathrm{w}} := \tau_{\mathrm{w}}^{v,c} := \inf\{t \geq s : \ L_t > 0\}.$$

STEP 1: GENERIC CROSSING LEMMA

**Lemma A.16** (Crossing once width is below the advantage). *Let $\mathcal{G} := \{\forall t \geq s : \ L_t \leq \bar{\delta}_t \leq U_t\}$. Assume the weak alternative $H_{\mathrm{w},1}(\Delta)$ holds for some $\Delta > 0$ and $t_0 \geq s$. Define the CS width $w_t := U_t - L_t$. Then on $\mathcal{G}$, for all $t \geq t_0$,*

$$L_t \geq \Delta - w_t.$$

*In particular, on $\mathcal{G}$,*

$$\tau^{\mathrm{s}} \leq \inf\{t \geq t_0 : \ w_t < \Delta\}.$$

*Consequently, if $w_t \to 0$ on $\mathcal{G}$, then $\tau^{\mathrm{s}} < \infty$ on $\mathcal{G}$.*

*Proof.* Work on $\mathcal{G}$. For $t \geq t_0$, the weak alternative gives $\bar{\delta}_t \geq \Delta$. Since $\bar{\delta}_t \leq U_t$ and $L_t \leq \bar{\delta}_t$ on $\mathcal{G}$,

$$L_t = U_t - (U_t - L_t) \geq \bar{\delta}_t - w_t \geq \Delta - w_t.$$

If $w_t < \Delta$, then $L_t > 0$ and therefore $\tau^{\mathrm{s}} \leq t$. The remaining statements follow immediately. $\square$

Using (20), we immediately obtain:

$$\mathbb{P}(\tau^{\mathrm{s}} < \infty) \geq \mathbb{P}(\mathcal{G}) \geq 1 - \alpha^{v,c}.$$

STEP 2: OUR EXPLICIT EMPIRICAL BERNSTEIN CS

We now *directly define* the empirical Bernstein CS used in our paper from (Choe & Ramdas, 2024), in the exact form needed for the proof.

**Empirical Bernstein radius (time-uniform).** Assume $\Delta_t \in [-1, 1]$ a.s. Define the predictable empirical variance proxy

$$\hat{V}_t := \frac{1}{t - s + 1} \sum_{u=s}^{t} (\Delta_u - \bar{\Delta}_{u-1})^2, \qquad \bar{\Delta}_{s-1} := 0.$$

Let

$$\ell_n(\alpha) := \log \frac{1}{\alpha} + \log \log(en), \qquad n \geq 1,$$

and define the (two-sided) empirical Bernstein half-width

$$r_n(\alpha) := \sqrt{\frac{2 \hat{V}_t \, \ell_n(\alpha)}{n}} + \frac{3 \, \ell_n(\alpha)}{n}, \qquad n = t - s + 1. \tag{21}$$

(Any equivalent time-uniform empirical Bernstein CS from our construction can be used here; constants are universal.)

Define

$$L_t := \bar{\Delta}_t - r_{t-s+1}(\alpha^{v,c}), \qquad U_t := \bar{\Delta}_t + r_{t-s+1}(\alpha^{v,c}), \qquad t \geq s.$$

**Theorem A.17** (Time-uniform empirical Bernstein CS (our instantiation)). *For any $\alpha \in (0, 1)$,*

$$\mathbb{P}\Big(\forall t \geq s : \ \bar{\delta}_t \in [L_t, U_t]\Big) \geq 1 - \alpha.$$

**Width bound.** By construction,

$$w_t = U_t - L_t = 2 r_{t-s+1}(\alpha^{v,c}).$$

STEP 3: SOLVING EXPLICITLY FOR THE CROSSING TIME

By Lemma A.16, on $\mathcal{G}$ it suffices to have

$$w_t < \Delta \quad \Longleftrightarrow \quad 2r_n(\alpha^{v,c}) < \Delta, \qquad n = t - s + 1.$$

Using (21), a sufficient condition is

$$2\sqrt{\frac{2\hat{V}_t\,\ell_n(\alpha^{v,c})}{n}} + \frac{6\,\ell_n(\alpha^{v,c})}{n} < \Delta.$$

Since $\Delta_t \in [-1, 1]$, we have the crude bound $\hat{V}_t \leq 4$ a.s., so a deterministic sufficient condition is

$$2\sqrt{\frac{8\,\ell_n(\alpha^{v,c})}{n}} + \frac{6\,\ell_n(\alpha^{v,c})}{n} < \Delta. \tag{22}$$

We now solve (22).

**Controlling the square-root term.**    Require

$$2\sqrt{\frac{8\,\ell_n}{n}} \leq \frac{\Delta}{2} \quad \Longleftrightarrow \quad \sqrt{\frac{8\,\ell_n}{n}} \leq \frac{\Delta}{4} \quad \Longleftrightarrow \quad \frac{8\,\ell_n}{n} \leq \frac{\Delta^2}{16} \quad \Longleftrightarrow \quad n \geq \frac{128}{\Delta^2}\,\ell_n.$$

**Controlling the linear term.**    Also require

$$\frac{6\,\ell_n}{n} \leq \frac{\Delta}{2} \quad \Longleftrightarrow \quad n \geq \frac{12}{\Delta}\,\ell_n.$$

Since $\Delta \leq 1$, the $\Delta^{-2}$ requirement dominates asymptotically, so it is enough to enforce

$$n \;\geq\; \frac{C}{\Delta^2}\,\ell_n(\alpha^{v,c})$$

for a universal constant $C$.

**Self-consistency and the $\log\log$ term.**    Recall $\ell_n(\alpha) = \log(1/\alpha) + \log\log(en)$. If we pick

$$n \geq \frac{C}{\Delta^2}\left(\log\frac{1}{\alpha^{v,c}} + \log\log\frac{e}{\alpha^{v,c}}\right),$$

then $\log\log(en)$ is at most a constant times $\log\log(e/\alpha^{v,c})$ (up to additive constants), so the inequality is self-consistent. Therefore, there exists a universal constant $C' > 0$ such that on $\mathcal{G}$,

$$\tau^{\mathrm{s}} \leq t_0 - 1 + \frac{C'}{\Delta^2}\left(\log\frac{1}{\alpha^{v,c}} + \log\log\frac{e}{\alpha^{v,c}}\right),$$

i.e.,

$$\tau^{\mathrm{s}} - t_0 = \tilde{\mathcal{O}}\!\left(\frac{\log(1/\alpha^{v,c})}{\Delta^2}\right) \quad \text{on } \mathcal{G}.$$

A.6.3. FINAL CONCLUSION

*Proof of Theorem 4.2. (Wealth/betting, strong alternative).* Assume Assumption A.15. Part I shows that for any $\eta \in (0, 1)$, with probability at least $1 - \eta$,

$$\tau^{v,c} \leq s + n_0 + N_\eta - 1,$$

where

$$N_\eta = \left\lceil \frac{32}{\Delta^2}\left(\log\frac{1}{\alpha^{v,c}} + \log\frac{1}{w_\star} + n_0\log\frac{1}{1 - \beta_\star} + \log\frac{1}{\eta}\right)\right\rceil.$$

In particular $\tau^{v,c} < \infty$ almost surely.

*(CS-crossing, weak alternative).* Assume the weak alternative: there exist $\Delta > 0$ and $t_0 \geq s$ such that $\bar{\delta}_t \geq \Delta$ for all $t \geq t_0$. Let $(L_t, U_t)$ be the empirical Bernstein CS defined above. By Theorem A.17, the coverage event $\mathcal{G}$ holds with probability at least $1 - \alpha^{v,c}$. On $\mathcal{G}$, Lemma A.16 implies that the stopping time $\tau_{\mathrm{w}}^{v,c} = \inf\{t \geq s : L_t > 0\}$ is finite, and solving the width condition using (21) yields

$$\tau_{\mathrm{w}}^{v,c} - t_0 = \tilde{\mathcal{O}}\left(\frac{\log(1/\alpha^{v,c})}{\Delta^2}\right) \quad \text{with probability at least } 1 - \alpha^{v,c}.$$

This completes the proof of Theorem 4.2. $\qquad\qquad\qquad\qquad\qquad\qquad\qquad\qquad\qquad\qquad\qquad\qquad\quad$ $\square$

## B. Implementation Details and Parameters

This appendix provides implementation-specific details underlying Algorithm 1.

### B.1. Discrete-Mixture Betting Strategy

In theory, the betting-based test can be expressed as a continuous mixture over betting fractions,

$$W_t = \int_0^1 \prod_{i=s}^t \left(1 + \beta\,\Delta_i\right) d\pi(\beta), \tag{23}$$

where $\pi$ denotes the Jeffreys prior on $[0, 1]$. In practice, we have to resort to approximation. We use a finite discrete mixture, which eliminates approximation error entirely, and allow to maintain the theoretical guarantees

**Discrete-mixture wealth.** Fix a finite grid of betting fractions

$$\mathcal{B} := \{\beta_1, \ldots, \beta_K\} \subset [0, 1],$$

for example a geometric or uniform grid such as $\mathcal{B} = \{0, 0.01, 0.02, \ldots, 0.99\}$. Let $\{w_k\}_{k=1}^K$ be nonnegative prior weights satisfying $\sum_{k=1}^K w_k = 1$.

The betting wealth is defined as

$$W_t := \sum_{k=1}^K w_k \prod_{i=s}^t \left(1 + \beta_k\,\Delta_i\right). \tag{24}$$

For each $k$, the process $W_t^{(k)} := \prod_{i=s}^t (1 + \beta_k \Delta_i)$ is a nonnegative supermartingale under the null hypothesis $\mathbb{E}[\Delta_t \mid \mathcal{F}_{t-1}] \leq 0$. Therefore, $W_t$ is itself a nonnegative supermartingale.

**Validity and efficiency.** This construction has three key consequences: (i) *Exact anytime-validity*: a finite sum of supermartingales remains a supermartingale, so Ville's inequality applies directly; (ii) *Approximation of the continuous Universal Portfolio*: standard online portfolio results imply that a sufficiently fine grid approximates the continuous mixture up to an additive $O(\log K)$ term in log-wealth; and (iii) *computational efficiency*: updating $K$ grid points incurs only $O(K)$ cost per step and is typically faster than Monte Carlo integration. We use $K = 100$.

Unless stated otherwise, all theoretical guarantees in the main text apply exactly to the discrete-mixture wealth (24).

### B.2. Significance-Level Allocation

We allocate significance levels only to tests that are actually instantiated. Let $\mathcal{T}$ denote the random countable set of node–candidate tests ever created by the algorithm. It is enough for global validity that, pathwise,

$$\sum_{(v,c)\in\mathcal{T}} \alpha^{v,c} \leq \alpha.$$

One can use the following online alpha-spending rule. For $d \geq 0$, $m \geq 1$, and $q \geq 1$, define

$$a_d = \frac{6}{\pi^2(d+1)^2}, \qquad b_m = \frac{6}{\pi^2 m^2}, \qquad g_q = \frac{6}{\pi^2 q^2}.$$

Thus $\sum_{d\geq 0} a_d = \sum_{m\geq 1} b_m = \sum_{q\geq 1} g_q = 1$.

For a node $v$, let $d(v)$ be its depth and let $r(v)$ be its creation rank among nodes of the same depth. Candidate splits may be generated adaptively. Let $q$ index the candidate-generation calls made at node $v$. On the $q$-th call, after removing duplicate or previously tested candidates, suppose $B_{v,q}$ new candidates remain. If $B_{v,q} = 0$, no level is spent. Otherwise, every new candidate $c$ in that batch receives

$$\boxed{\alpha^{v,c} = \alpha_{\text{tree}}\, a_{d(v)}\, b_{r(v)}\, g_q\, \frac{1}{B_{v,q}}}.$$

For a single tree, take $\alpha_{\text{tree}} = \alpha$. For a forest with $M$ trees, take $\alpha_{\text{tree}} = \alpha/M$.

The level $\alpha^{v,c}$ is fixed before the first observation used to update the corresponding test statistic. Hence it is predictable at the test start time.

To verify summability, fix any realized run of the algorithm. Let $\mathcal{V}$ be the set of nodes ever created and let $\mathcal{C}_{v,q}$ be the set of new candidates instantiated at node $v$ on proposal call $q$. Then

$$\sum_{(v,c)\in\mathcal{T}} \alpha^{v,c} = \alpha_{\text{tree}} \sum_{v\in\mathcal{V}} a_{d(v)} b_{r(v)} \sum_{q\geq 1} g_q \sum_{c\in\mathcal{C}_{v,q}} \frac{1}{B_{v,q}}$$

$$\leq \alpha_{\text{tree}} \sum_{d\geq 0} a_d \sum_{m\geq 1} b_m \sum_{q\geq 1} g_q = \alpha_{\text{tree}}.$$

Therefore

$$\sum_{(v,c)\in\mathcal{T}} \alpha^{v,c} \leq \alpha_{\text{tree}}$$

pathwise.

## B.3. Candidate Split Generation

To avoid biasing the comparison, candidate splits are generated using exactly the same mechanisms as in the baseline Hoeffding Tree. Depending on the configuration, each node employs either a Gaussian splitter or a histogram-based splitter, maintaining feature-wise sufficient statistics. By default, HT uses the gaussien splitter. When queried, the node proposes a finite set of promising candidates based on empirical impurity reduction. In our experiments, we use the default setting of proposing up to 10 candidates per time step; however, we automatically discard duplicates and ensure that previously proposed splits are not re-evaluated.

Overall, we use exactly the same configuration as the default Hoeffding Tree. Our method only differs from the baseline only in the statistical test used to commit splits.

## B.4. Empirical Bernstein Confidence Sequence

For the confidence-sequence–based test, we use a time-uniform empirical Bernstein confidence sequence as in (Choe & Ramdas, 2024). Assume $\Delta_t \in [-1,1]$ almost surely and let $s$ denote the test start time.

**Empirical Bernstein radius (time-uniform).** Define the running average

$$\bar{\Delta}_t := \frac{1}{t-s+1} \sum_{u=s}^{t} \Delta_u, \qquad t \geq s,$$

and the predictable empirical variance proxy

$$\hat{V}_t := \frac{1}{t-s+1} \sum_{u=s}^{t} \left(\Delta_u - \bar{\Delta}_{u-1}\right)^2, \qquad \bar{\Delta}_{s-1} := 0.$$

Let

$$\ell_n(\alpha) := \log\frac{1}{\alpha} + \log\log(en), \qquad n \geq 1,$$

and define the empirical Bernstein half-width

$$r_n(\alpha) := \sqrt{\frac{2\,\hat{V}_t\,\ell_n(\alpha)}{n}} + \frac{3\,\ell_n(\alpha)}{n}, \qquad n = t - s + 1. \tag{25}$$

The resulting confidence sequence is

$$L_t := \bar{\Delta}_t - r_{t-s+1}(\alpha^{v,c}), \qquad U_t := \bar{\Delta}_t + r_{t-s+1}(\alpha^{v,c}), \qquad t \geq s.$$

# C. Additional experiments

This section presents supplementary experiments that further characterize the empirical behavior of our anytime-valid decision tree framework. We first isolate and compare the proposed betting-based and confidence-sequence–based split-certification mechanisms in Appendix C.1, and then report additional comparisons against established streaming ensemble baselines in Appendix C.2.

## C.1. Comparison between $\text{AVT}_\text{B}$ and $\text{AVT}_\text{CS}$

We compare $\text{AVT}_\text{B}$ and $\text{AVT}_\text{CS}$ across a diverse collection of real-world regression and classification data streams. The two methods differ only in the underlying anytime-valid testing mechanism used to certify splits betting-based tests for $\text{AVT}_\text{B}$ versus confidence-sequence–based tests for $\text{AVT}_\text{CS}$, allowing for a clean empirical comparison of their practical behavior. Figure 3 reports full prequential and distribution performance trajectories on twelve benchmark datasets.

Overall, $\text{AVT}_\text{CS}$ underperforms $\text{AVT}_\text{B}$, though it remains competitive with and typically outperforms—the ARF baseline.

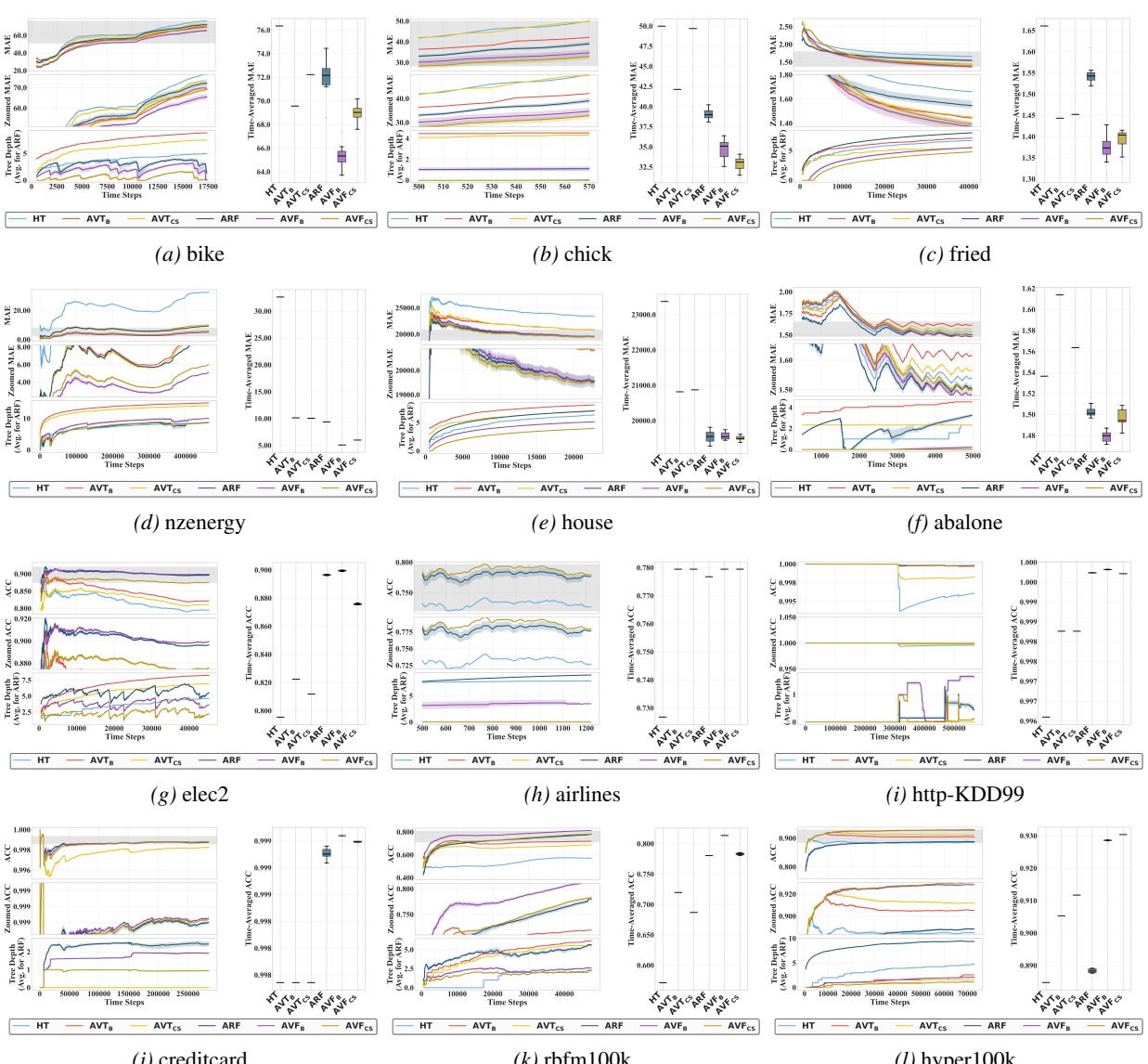

*(a)* bike      *(b)* chick      *(c)* fried

*(d)* nzenergy      *(e)* house      *(f)* abalone

*(g)* elec2      *(h)* airlines      *(i)* http-KDD99

*(j)* creditcard      *(k)* rbfm100k      *(l)* hyper100k

*Figure 3.* Performance comparison between $\text{AVT}_\text{B}$ and $\text{AVT}_\text{CS}$ across regression datasets **(a)–(f)** and classification datasets **(g)–(l)**.

## C.2. Other baselines

In this section, we report additional comparisons against existing streaming ensemble baselines. We use the packages river (Montiel et al., 2021) and CapyMOA (Gomes et al., 2025) for the baselines. For classification, we use SOKNL (Sun et al., 2022), OnlineSmoothBoost (Chen et al., 2012), StreamingRandomPatches (Gomes et al., 2019), OzaBoost (Oza & Russell, 2001), and ARF. For regression, we compare only against SOKNL and ARF, as the previous methods are not currently available for regression tasks.

As noted by (Read & Zliobaite, 2025), the streaming literature has predominantly focused on classification, with regression receiving comparatively limited attention until recently. We also exclude SGBT (Gunasekara et al., 2025) from our evaluation due to unresolved implementation issues that prevent reliable use; details of the issues are provided in Appendix D.

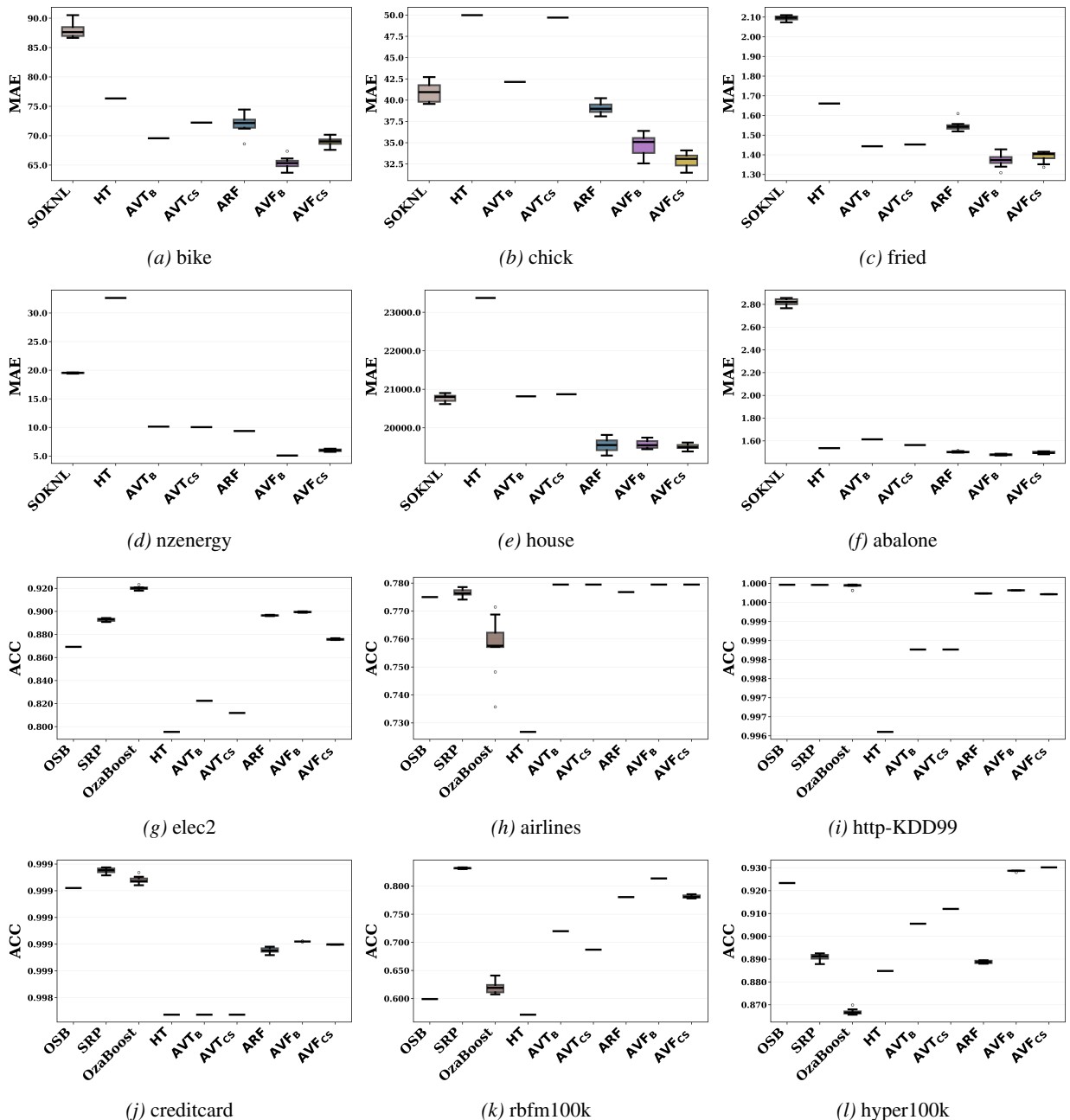

*Figure 4.* Regression datasets **(a)–(f)** and classification datasets **(g)–(l)**.

# D. Implementation issues with baselines

During our evaluation, we encountered several implementation issues with existing baselines, most notably with the Streaming Gradient Boosting Tree (SGBT) (Gunasekara et al., 2025).

**Degenerate regression behavior.** When used for regression, SGBT consistently outputs a zero prediction throughout the entire stream, independently of the dataset or training progress. This behavior is illustrated by the diagnostic experiment shown in Fig. 5.

```python
import numpy as np
from capymoa.datasets import Fried, Bike
from capymoa.regressor import StreamingGradientBoostedRegression

def check_regression_predictions(stream, name):
    reg = StreamingGradientBoostedRegression(schema=stream.get_schema())
    preds = []

    while stream.has_more_instances():
        inst = stream.next_instance()
        yhat = reg.predict(inst)
        if yhat is not None:
            preds.append(float(yhat))
        reg.train(inst)

    print(f"{name}: unique predictions =", np.unique(preds))

check_regression_predictions(Fried(), "Fried")
check_regression_predictions(Bike(), "Bike")

✓ [1] 37s 437ms

Fried: unique predictions = [0.]
Bike: unique predictions = [0.]
```

*Figure 5.* SGBT regression behavior: the model outputs a constant zero prediction over the full stream.

**Source inspection.** Inspection of the source code reveals that the SGBT regressor inherits from a classifier-oriented base class (Fig. 6), which may implicitly enforce classification-specific behavior. We attempted several fixes but were unable to obtain a functioning regression variant.

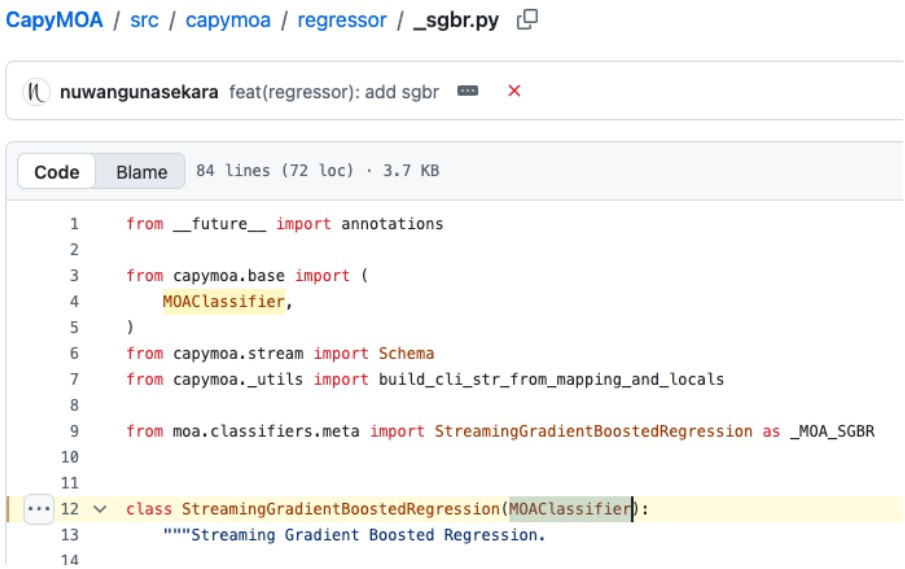

*Figure 6.* SGBT class hierarchy highlighting inheritance from a classifier base.

**Stability issues.** More broadly, several CapyMOA baselines exhibit recurrent runtime failures. For example, experiments on several datasets terminate with Java errors, which we were unable to resolve. All experiments were run on AWS EC2 instances. The main evaluation used a c7i.16xlarge instance (64 vCPUs, 128 GiB RAM), and we also attempted a memory-optimized r8a.16xlarge configuration (64 vCPUs, 512 GiB RAM), but the stability issues persisted.

```
java.lang.java.lang.OutOfMemoryError: java.lang.OutOfMemoryError: Java heap space

The above exception was the direct cause of the following exception:

Traceback (most recent call last):
  File "AbstractClassifier.java", line 101, in moa.classifiers.AbstractClassifier.getVotesForInstance
Exception: Java Exception

The above exception was the direct cause of the following exception:
```

*Figure 7.* Example Java runtime error encountered when running CapyMOA baselines.

# E. Experiments details

All experiments were run on AWS EC2 instances. The main evaluation used a c7i.16xlarge instance (64 vCPUs, 128 GiB RAM).

