# OpenReview forum: "Correcting Split Selection in Online Decision Trees via Anytime-Valid Inference"
_ICML.cc/2026/Conference — ICML 2026 spotlight_

### Official Review · Reviewer_AhFs · 2026-02-27

**Soundness:** 4
**Presentation:** 4
**Significance:** 3
**Originality:** 3
**Overall Recommendation:** 5
**Confidence:** 2

**Summary:**

This paper introduces new splitting criterions for decision trees when learning from data streams. The splitting criterions are based on the safe, anytime-valid inference framework. This work addresses the issues of Hoeffding trees using standard concentration inequalities which assume fixed sample size. Thus, the probability that Hoeffding trees makes a false split at *some* point goes up over time.

In contrast to this, the new splitting rules presented in this work bounds the probability of false splits over all time steps simultaneously. The splitting rules work by deciding a candidate split when a node is created. In subsequent time-steps they compare the candidate split to no split using the testing-by-betting framework, ensuring splits are only made when they are confident the candidate is not a false split.

Beyond preventing false splits they also give bounds on how long one has to wait until a split with advantage $\Delta$ will be committed by the splitting rule. Furthermore, they show that the expected loss is monotonically decreasing across commits.

Lastly, the paper includes experiments which compare their new tree building algorithm with Hoeffding trees both when a single decision tree and when using Adaptive Random Forests. Through these, they provide empirical evidence that their tree building strategy outperforms Hoeffding trees in most cases.

**Compliance With Llm Reviewing Policy:**

Affirmed.

**Final Justification:**

My major point in my review was about the reason for why minimizing false splits are undesirable and how we know Hoeffding trees are indeed suboptimal in that regard. After reading the rebuttal and looking at the relevant parts of the paper again, I realize that these concerns were not an issue. The example provided was helpful to my own understanding, but I also acknowledge that the average reader might not need such detailed explanation for the motivation. As a result, I have updated my score.

The rest of my points were more minor, and they were all addressed appropriately.

**Key Questions For Authors:**

I have 4 questions.

1. Regarding first paragraph in **weaknesses** section above, do you have such an example?
2. See second paragraph of **weaknesses**.
3. Bottom left of page 7, you say it is expected that $\text{AVT}_\text{B}$ grows deeper than HT. Your reasoning refers to depth being necessary to match accuracy of an ensemble. Can you elaborate on this? HT is not an ensemble, and isn't this the model you are comparing the depth to?
4. You assume $C_v$ to be finite. Is this an artifact of your proofs, or is this a neccessary condition to be able to get results of this nature?

**Limitations:**

yes

**Strengths And Weaknesses:**

**Strengths:**\
The paper is well written and easy to follow. They outline their contributions early on and motivate their work well already in the very beginning. Also, the authors make sure to introduce the concepts they use and don't assume too much prior knowledge of the reader.
Regarding soundness, the paper provides proofs of all their claims (although I have not read the proofs) and have even done experiments to back up the applicability of their theory.
I think the issues the authors present with the current state of theory for Hoeffding trees is very relevant, and it seems very natural to introduce this different splitting rule for decision trees.

**Weaknesses:**\
One thing I feel is missing would be a concrete synthetic example of when Hoeffding trees are provably inferior to the proposed anytime-valid trees. As I understand the introduction of the paper, the main concern of Hoeffding trees is that the *current* approaches don't give guarantees across all time steps. However, this doesn't rule out that Hoeffding trees *could* have such properties. An example proving that Hoeffding trees can never hope to get such guarantees would in my opinion improve the contribution.

Furthermore, I am missing a discussion about why false splits are undesirable. Do false splits provably imply worse performance, or is it just bad from an intuitive perspective? And does this change when we do random forests?

Also, the paper never clearly states what a "false split" is. I think (even though most reader probably would be able to infer the definition themself) it would be appropriate to define it explicitly, since the term is used in one of the main theorems of the paper.

Moreover, I have some concrete minor comments which should be easy to address:

- 085(left): $v_L^c$ should be $v_c^L$ and the same goes for the right child.
- 105(left): there is a "|" which shouldn't be there.
- 167,213,227: Names of citation should not be in parenthesis when used in the sentence. I didn't explicitly look for this, so there may be more occurrences of this mistake.
- 182(right): Equation flows into margin.
- It would be nice, if it was explicitly mentioned which statements in the appendix implies Theorem 4.1. At the moment, Theorem 4.1 is not mentioned in the appendix.

---

> ### Author Rebuttal · Authors · 2026-03-30
>
> We thank the reviewer for the careful reading and positive assessment of the paper’s motivation, clarity, and technical soundness. We address the main questions below.
>
> > **Q1. An example proving that Hoeffding trees can never hope to get such guarantees would, in my opinion, improve the contribution. Furthermore, I am missing a discussion about why false splits are undesirable. Do false splits provably imply worse performance...and does it impact random forests?**
>
> The core issue is that standard HT split rules repeatedly monitor a fixed-time Hoeffding/McDiarmid threshold and stop at the first crossing. This is an optional stopping setting: fixed-time concentration bounds are not valid under continuous sequential monitoring. Thus, even under the null (when no split is truly beneficial), the probability of eventually crossing the threshold can become arbitrarily large over time due purely to stochastic fluctuations. See Fig. 1 in Howard et al. (2021), cited in line 36: a fixed-time confidence interval for a zero mean can eventually exclude zero with high probability if monitored indefinitely.
>
> A simple null example makes this concrete. Suppose a leaf receives i.i.d. samples with $Y_t \sim \mathrm{Bernoulli}(1/2)$, $X_t \sim \mathrm{Bernoulli}(1/2)$, and $X_t \perp Y_t$. Then $X_t$ is uninformative, so the correct decision is **never to split**. Under repeated monitoring of a fixed-time threshold, however, one can still eventually split due to noise alone. In contrast, AVT controls this event at all times by construction.
>
> False splits can also hurt performance. In the null setting above, under Brier loss, the prediction at a leaf is the empirical mean $\hat p$, and the risk is $\mathbb{E}(Y-\hat p)^2 = \tfrac14 + \mathrm{Var}(\hat p)$. Thus, for an unsplit leaf with $n$ samples, $R_{\mathrm{unsplit}}(n) = \frac14 + \frac{1}{4n}$, while a false split into children of sizes $n_L,n_R$ gives $R_{\mathrm{split}}(n_L,n_R) = \frac14 + \frac18\left(\frac1{n_L}+\frac1{n_R}\right)$. Therefore, $R_{\mathrm{split}} - R_{\mathrm{unsplit}} \ge \frac{1}{4n} > 0$. So a false split strictly increases expected risk: it fragments the data, increasing variance without reducing bias.
>
> This is also consistent with the warm-up experiment in Section 4.4. There, the data are fully i.i.d. and stationary, so the observed behaviour cannot be attributed to drift. Yet Figure 1 shows abrupt drops in generalisation performance for HT, whereas the anytime-valid variants remain much more stable. However, note that with enough data, the estimation variance may eventually vanish, as we observe in Figure 1. Rather, the issue is that HT can make such splits too early and without valid sequential guarantees, creating unnecessary instability that AVT avoids.
>
> Finally, early splits, especially near the root, are known to have a disproportionate impact on the final tree, making incorrect splits particularly harmful (see e.g., Near-Optimal Decision Trees in a SPLIT Second). This further supports a more conservative split-selection rule, such as anytime-valid testing.
>
> For random forest, averaging can partially mitigate the effect of any single false split, but the issue does not disappear. Spurious splits can still weaken individual base learners, and our results suggest that replacing HT with AVT within ARF yields ensembles that are both more accurate and smaller.
>
> ---
>
> > **Q2. Clarification on Tree Depth (AVT vs. HT vs. AVF)**
>
> We apologise for the confusion on page 7. The intended comparison was between the standalone tree (**AVT**) and its ensemble version (**AVF**), not between AVT and HT.
>
> The intuition is as follows. For a single tree, depth is the main mechanism for increasing representational capacity, so a statistically cautious learner such as AVT may still eventually grow deeper if the task requires more complex partitions. In contrast, in an ensemble such as AVF, complexity can be achieved by aggregating many randomised trees, reducing the pressure on any individual tree to grow very deep. We will clarify this explicitly in the final version.
>
> ---
>
> > **Q3. The Assumption of Finite (C_v)**
>
> The assumption that the number of candidate splits (C_v) is finite reflects the standard practical setting rather than a restrictive theoretical artefact. In standard tree implementations (e.g., CART), the number of candidate splits is finite because there are finitely many features and, for a given dataset, finitely many possible split thresholds. Our results, therefore, apply under the same finite-candidate regime used in standard practical implementations.
>
> ---
>
> > **Minor Corrections and Typos**
>
> We thank the reviewer for the careful proofreading. We will correct the indices for (v_L^c), the stray “(|)”, the citation formatting, and the equation overflow. We will also explicitly point to Theorem 4.1 and Theorem A.2 in the appendix for ease of reference and clarify that a false split is one made when the null hypothesis is true.

---

> > ### Author Rebuttal · Reviewer_AhFs · 2026-03-31
> >
> > The authors have addressed my points satisfactorily. The example was very helpful for my own understanding, but I can see why this might be unnecessarily elaborate if included in the paper. I have updated my score to reflect my current recommendation.

---

### Official Review · Reviewer_hDQ9 · 2026-03-08

**Soundness:** 3
**Presentation:** 3
**Significance:** 3
**Originality:** 3
**Overall Recommendation:** 5
**Confidence:** 1

**Summary:**

The paper identifies an important statistical issue in existing Hoeffding-tree analyses: optional stopping. that the statistical justification commonly used in Hoeffding Trees is invalid because fixed-time concentration bounds are being used together with data-dependent stopping. The authors propose replacing this with anytime-valid inference, so that split decisions remain valid under optional stopping and even under non-stationary or dependent streams.

The theoretical claims are threefold. First, with an appropriate allocation of per-test significance levels, the procedure gives anytime-valid global control of false splits over the lifetime of the tree. Second, under a persistent predictive advantage, the stopping time is finite. Third, under stationary i.i.d. data and convex loss, the expected risk of the deployed tree is monotone between commits, and it also improves at commit times under the weak-test formulation with an advantage threshold.

Empirically, the paper evaluates standalone trees and adaptive random forests on a synthetic stationary benchmark and on 12 real or synthetic data streams. The reported results suggest that the anytime-valid variants are more stable than standard Hoeffding Trees, and that the forest version often achieves better predictive performance with smaller trees, though with noticeably higher update costs.

**Compliance With Llm Reviewing Policy:**

Affirmed.

**Final Justification:**

The rebuttal resolved my main concern regarding when the selected challenger approximates the classical HT best split. I therefore increase my score by one point.

**Key Questions For Authors:**

Since the method compares each candidate split to the unsplit leaf, rather than directly comparing candidate splits to one another, under what conditions should the selected challenger approximate the best split in the classical HT sense?

**Limitations:**

yes

**Strengths And Weaknesses:**

Strengths
1. The identification of the problem is novel and obeys the general instinct of split settings in statistics.
2. The anytime-valid reformulation is technically well aligned with that problem, and the main theorems give clear guarantees for global false-split control and finite commitment under persistent advantage.



Weaknesses
1. The practical significance is moderated by runtime overhead. The paper reports that inference time is similar, but update time can increase substantially.

---

> ### Author Rebuttal · Authors · 2026-03-30
>
> We thank the reviewer for the positive assessment of the paper’s motivation and theoretical and empirical contributions.
>
> > **Q.** **Since the method compares each candidate split to the unsplit leaf, rather than directly comparing candidate splits to one another, under what conditions should the selected challenger approximate the best split in the classical HT sense?**
>
>
> Our current procedure should approximate the classical HT choice when there is a clear and persistent separation among candidates at the population level: namely, when one split has a strictly larger expected improvement over the unsplit leaf than all others, and this gap is large enough to be detected earlier than competing candidates. In that regime, the statistically strongest challenger is also the one most likely to be certified first, so the selected split should align with the classical best split.
>
> If exact best-split identification is desired, the framework can be extended to include additional anytime-valid pairwise comparisons of candidate splits. For example, one could commit only when the upper confidence sequence bound for the selected candidate’s loss is below the lower confidence bounds of all competitors, or equivalently, when pairwise anytime-valid tests certify that no alternative candidate has a smaller loss.

---

> > ### Author Rebuttal · Reviewer_hDQ9 · 2026-04-05
> >
> > The rebuttal addressed my concern by clarifying the conditions under which the selected challenger should approximate the classical HT best split.

---

### Official Review · Reviewer_t3xp · 2026-03-13

**Soundness:** 3
**Presentation:** 3
**Significance:** 3
**Originality:** 3
**Overall Recommendation:** 5
**Confidence:** 4

**Summary:**

The paper present a novel construction of decision tree when the training set is given in an on line fashion. The proposed methodology is shown to improve with respect to the Hoeffding trees. First and foremost for the new construction the statistical guarantees are valid also under data-dependet stopping rule and in the case of non-stationary data. Moreover, the experiments show that the also the prediction capability of the new AVT compares favourably with the state of the art both as stand alon tree and when used  in adaptive random forests.

**Compliance With Llm Reviewing Policy:**

Affirmed.

**Key Questions For Authors:**

It would be nice to see experiments of implementations exploiting the parallelizability of the best split choice.

**Limitations:**

yes

**Strengths And Weaknesses:**

Strengths: the theoretical analysis proves validity beyond the (restrictive and not always satisfied) assumptions needed for the analogous results for state of the art online decision trees; moreover, the experimental analysis shows that both rules proposed for the splitting criterion yield valid predictors, mostly outperforming HT; the smaller size of the trees is also interesting, with respect to the typical explainability property of decision trees.
Weakness: the splitting rules appears to be generally more time consuming. However the prediction time remains competitive, thanks to the smaller size of the trees
Originality: as far as I known the approach, as employed in the the context of Online DT is new
Significance: both the theoretical result and the experimental validation make the proposed solution provide clear advancement in the area.

---

> ### Author Rebuttal · Authors · 2026-03-30
>
> We thank the reviewer for the positive assessment of the paper’s originality, significance, and experimental and theoretical contributions.
>
> > **Q.** **It would be nice to see experiments of implementations exploiting the parallelizability of the best split choice.**
>
> We agree this is an important practical point. We implemented a parallelized version of **AVT** and benchmarked it against both the original AVT and standard HT. The preliminary results below report per-sample update time (ms) over 20k samples.
>
> | Dataset | Model | Performance | Update (ms) | Speedup |
> |---|---|---:|---:|---:|
> | **RandomRBF** | Standard HT | Acc = 0.763 | 0.04 | — |
> |  | AVT | Acc = **0.831** | 3.96 | 1.0× |
> |  | AVT (parallel) | Acc = **0.831** | 1.17 | **3.4×** |
> | **Hyperplane** | Standard HT | Acc = 0.864 | 0.03 | — |
> |  | AVT | Acc = **0.920** | 5.46 | 1.0× |
> |  | AVT (parallel) | Acc = **0.920** | 0.86 | **6.4×** |
> | **Friedman** | Standard HT | MAE = 1.786 | 0.03 | — |
> |  | AVT | MAE = **1.580** | 2.81 | 1.0× |
> |  | AVT (parallel) | MAE = **1.580** | 1.39 | **2.0×** |
>
> These results show that parallelisation substantially reduces the training overhead, yielding roughly **2× to 6× faster updates**, while preserving the predictive gains of AVT.
>
> We will include this discussion in the revised version and report parallel speedups across the full benchmark suite.

---

> > ### Author Rebuttal · Reviewer_t3xp · 2026-04-02
> >
> > The authors comprehensively answered my question

---

### Decision · Program_Chairs · 2026-04-30

**Decision:**

Accept (spotlight)

**Comment:**

This work addresses the problem of tree-splitting in data stream learning, where existing approaches rely on heuristics and cannot properly control the error rate of incorrect splits. To overcome this limitation, the authors propose a new learning approach based on anytime-valid inference, which enables controlled split decisions at arbitrary times. The method provides valid statistical guarantees even in non-stationary environments, while improving predictive performance and producing simpler models. All reviewers recognize the novelty and significance of the proposed approach, and the work represents a strong contribution to statistically grounded data stream learning.